# The role of ER exit sites in maintaining P-body organization and integrity during *Drosophila melanogaster* oogenesis

Samantha N Milano [ID] [1,2], Livia V Bayer[1], Julie J Ko[1], Caroline E Casella [ID] [1] & Diana P Bratu [ID] [1,2] ✉

## Abstract

**Processing bodies (P-bodies) are cytoplasmic membrane-less organelles which host multiple mRNA processing events. While the fundamental principles of P-body organization are beginning to be elucidated in vitro, a nuanced understanding of how their assembly is regulated in vivo remains elusive. Here, we investigate the potential link between ER exit sites and P-bodies in *Drosophila melanogaster* egg chambers. Employing a combination of live and super-resolution imaging, we find that P-bodies associated with ER exit sites are larger and less mobile than cytoplasmic P-bodies, indicating that they constitute a distinct class of P-bodies. Moreover, we demonstrate that altering the composition of ER exit sites has differential effects on core P-body proteins (Me31B, Cup, and Trailer Hitch), suggesting a potential role for ER exit sites in P-body organization. Furthermore, we show that in the absence of ER exit sites, P-body integrity is compromised and the stability and translational repression efficiency of the maternal mRNA, *oskar*, are reduced. Together, our data highlights the crucial role of ER exit sites in governing P-body organization.**

**Keywords** P-bodies; ER Exit Sites; Oogenesis; Post-transcriptional Regulation; Biological Condensates
**Subject Categories** Development; Organelles; RNA Biology

## Introduction

Investigations of biological condensates are unlocking insights into a broad array of fields ranging from mRNA regulation and stress response to long-distance transcript localization (Bayer et al, 2024; Eulalio et al, 2007a; Lyon et al, 2021). While the breadth of condensate function is beginning to be elucidated, little is known about how condensate assembly is regulated in vivo. In vitro studies have revealed that condensates can form de novo, via liquid–liquid phase separation (LLPS) (Burke et al, 2015; Elbaum-Garfinkle et al, 2015; Molliex et al, 2015; Nott et al, 2015). Such condensation

events are driven by weak mRNA:mRNA, mRNA:protein, and protein:protein interactions (Brangwynne et al, 2009; Van Treeck and Parker, 2018). Many proteins capable of LLPS have intrinsically disordered regions that allow for multiplexing interactions (Borcherds et al, 2021; Schütz et al, 2017). These weak interactions generate a local concentration of mRNAs and proteins which is significantly higher than the surrounding cytoplasm. Due to this disparity, it becomes thermodynamically favorable for a granule to form a distinct liquid phase (Banani et al, 2017; Brangwynne et al, 2009). Once formed, these liquid-like granules can mature and become solid-like as intra-condensate interactions strengthen (Garaizar et al, 2022; Lin et al, 2015). Whether this bottom-up mechanism is the only driving factor that dictates condensate formation in complex tissues is yet to be deciphered. Here, we look to highlight the influence of specialized regions of the ER, ER exit sites (ERES), on condensate formation in the *Drosophila melanogaster* female germline.

The *D. melanogaster* ovary is a complex tissue consisting of 16 to 20 chains of egg chambers called ovarioles. The germline of each egg chamber is composed of a 16-cell cyst inter-connected by cytoplasmic bridges called ring canals, allowing for a shared cytoplasm. Early in development, one of these cells becomes the oocyte, while the other 15 become nurse cells. The oocyte is transcriptionally silent throughout oogenesis while the nurse cells produce all the transcripts required for the oocyte's development (McLaughlin and Bratu, 2015). These transcripts must be stably maintained and translationally repressed as they are spatio-temporally localized to the distant oocyte (Lasko, 2012). One method by which this silenced transport may be achieved is via processing bodies (P-bodies) (Bayer et al, 2024; Weil et al, 2012).

P-bodies play an active role in the mRNA life cycle, facilitating transcript storage, translational repression, and mRNA decay (Eulalio et al, 2007a; Sheth and Parker, 2003; Standart and Weil, 2018). Unlike stress granules, which form in response to cellular stress, P-bodies are constitutively present in the cytoplasm (Protter and Parker, 2016; Standart and Weil, 2018). Notably, the number of P-bodies varies by cell type, hinting at their functional importance. For instance, oocytes and neurons both contain a prodigious number of P-bodies. As these are both polarized cells where transport of stable and translationally repressed transcripts occurs over long distances, it suggests that P-bodies may play a critical role

---

[1]Department of Biological Sciences, Hunter College, City University of New York, New York, NY 10065, USA. [2]Program in Molecular, Cellular, and Developmental Biology, The Graduate Center, City University of New York, New York, NY 10016, USA. ✉E-mail: bratu@genectr.hunter.cuny.edu

in this transport process (Lin et al, 2008; Zeitelhofer et al, 2008). Interestingly, these cell types also utilize localized ERES on the ER for proper protein localization.

ERES are specialized regions of the ER where COPII vesicles form, allowing for protein secretion (Kurokawa and Nakano, 2019). These sites are not always stochastically located along the ER; in some cell types, their distribution can be spatio-temporally regulated (Aridor et al, 2004). For example, increased ERES markers have been observed preferentially at the end of axons (Aridor and Fish, 2009). Similarly, in the *D. melanogaster* egg chamber, protein localization is regulated in part via selection of localized ERES (Herpers and Rabouille, 2004).

The functional link between the ER and P-bodies is not novel (Decker and Parker, 2006). Previous studies in *D. melanogaster* egg chambers have demonstrated that core P-body proteins, Me31B (RNA helicase), Cup (eIF4E -BP), and Trailer Hitch (Tral) (LSM 14-like) colocalize with the ER (Kugler et al, 2009; Wilhelm et al, 2005). These studies further revealed that Tral directly binds the mRNAs of two COPII proteins: Sar1 and Sec13, and in the absence of Tral protein, ERES are aberrantly localized, and secretion is disrupted indicating that P-bodies play a role in ERES regulation (Wilhelm et al, 2005). In *C. elegans*, the homolog of Tral, Car-1, similarly governs proper ER organization, implying that there may be a conserved connection between the ER and P-bodies (Squirrell et al, 2006). As evidence continues to mount for the spatio-temporal regulation of ERES by P-bodies, recent reports suggest the reciprocal regulation of P-bodies by the ER (Nguyen et al, 2024).

ER contact sites with membrane-bound organelles have previously been noted (Wu et al, 2018). Now, the possibility that the ER also forms contact sites with membrane-less organelles is emerging. In yeast, P-body components are directly associated with the ER, and blocking ER secretion leads to an increase in P-body formation at the ER (Kilchert et al, 2010). It has similarly been shown in *C. elegans* that ER sheets may contribute to protein condensation (Elaswad et al, 2024). Furthermore, in cell culture, the ER can directly facilitate P-body fission, bolstering the idea that the interface between the ER and P-bodies is conserved and has meaningful biological consequences (Lee et al, 2020).

Here, we look to further delineate the relationship between ERES and P-body assembly in vivo. Using super-resolution imaging and RNAi-facilitated knockdowns, we found that P-bodies at ERES are distinct from cytoplasmic P-bodies. Furthermore, upon knockdown of COPII components, we observed alterations in the morphology of key P-body protein condensates, suggesting a potential involvement of COPII proteins in overall P-body organization. Finally, in the absence of ERES, P-bodies did not form detectable condensates, and maternal mRNA, *oskar*, was prematurely expressed and degraded, a finding that could have implications for diseases ranging from cancer to neurodegeneration (Anderson et al, 2015; Wilby and Weil, 2023).

## Results

### P-bodies colocalized with ER exit sites are distinct from cytoplasmic P-bodies

P-bodies are heterogeneous in composition, size, and morphology (Eulalio et al, 2007a; Luo et al, 2018). Since P-bodies have been shown to interface with the ER (Kilchert et al, 2010), we sought to quantitatively characterize the distinctions between ER-associated P-bodies and cytoplasmic P-bodies. For our analysis, we chose to assess two condensate parameters: sphericity and volume as they provide insight into condensate state. Basic thermodynamics suggests that small volume, liquid-like droplets are spherical due to the influence of surface tension. Over time, condensates can coalesce, with a tendency to increase in volume in order to reduce interfacial tension. Liquid-like condensates can mature as stronger intra-condensate interactions develop making them more gel-like and less spherical over time (Lyon et al, 2021; Villegas et al, 2022).

As P-body characteristics may vary at different stages of development, we chose to only image nurse cells in stage 7 egg chambers to remain consistent throughout our analysis (Black square in Fig. 1A). We visualized P-bodies marked with endogenous Me31B-GFP and the ER by expressing UAS-KDEL-RFP induced with the Gal4/UAS system (Fig. 1B). These stocks were viable over multiple generations, suggesting that there are no catastrophic phenotypes associated with the transgene. To differentiate between cytoplasmic P-bodies and ER-associated P-bodies we used Imaris image analysis software to detect P-body (volume above 0.05 $\mu m^3$) and ER surfaces. We binned all detected P-bodies that have shortest distances to the ER ≤ 0 $\mu m$ (touch or overlap) as ER-associated P-bodies (Fig. 1B, full arrows) and any P-bodies with a shortest distance to the ER > 0 $\mu m$ (not touching) as a cytoplasmic P-body (Fig. 1B, arrowheads). Interestingly, we found that P-bodies associated with the ER were ~12% less spherical, ~80% larger (0.973 $\mu m^3$ compared to 0.196 $\mu m^3$), and had aspect ratios deviating further from 1 compared to their cytoplasmic counterparts (Figs. 1C,D and EV1A).

Next, we sought to determine the specific region of the ER where the difference in condensate characteristics occurs. We chose to focus on ERES-associated P-bodies because of the previous connection between P-bodies and ERES (Kugler et al, 2009; Wilhelm et al, 2005). We conducted the same analysis using Me31B-GFP and a Gal4-induced ERES protein, Sec31-RFP (Fig. 1E). To confirm that the transgene accurately reflected ERES localization, we verified its localization with a COPII antibody (Fig. EV1B). Interestingly, we found that ERES-associated P-bodies (Fig. 1E, full arrows) were ~25% less spherical, ~71% larger (0.324 $\mu m^3$ compared to 0.095 $\mu m^3$), and had aspect ratios farther from 1 when compared to cytoplasmic P-bodies (Fig. 1E, arrowheads), further suggesting that these P-bodies represent a distinct class of P-bodies (Figs. 1F,G and EV1C).

Interestingly, the average volume of ER-associated P-bodies is significantly larger than those associated with just ERES (0.973 $\mu m^3$ compared to 0.324 $\mu m^3$). This may be due to the differential localization of ERES and the whole ER membrane. ERES are centrally localized in the nurse cell cytoplasm while the ER membrane in general extends to regions of the nurse cell cytoplasm where P-bodies congregate and coalesce. For example, as the (-) ends of microtubules are oriented towards the periphery of the nurse cells, many P-bodies accumulate there (Bayer et al, 2024; Theurkauf and Hazelrigg, 1998). Similarly, we found the ring canal areas which allow for the cytoplasm to communicate between nurse cells, have an abundance of P-bodies while they attempt to move through a small volume, thus becoming bottlenecked and clustered into larger entities.

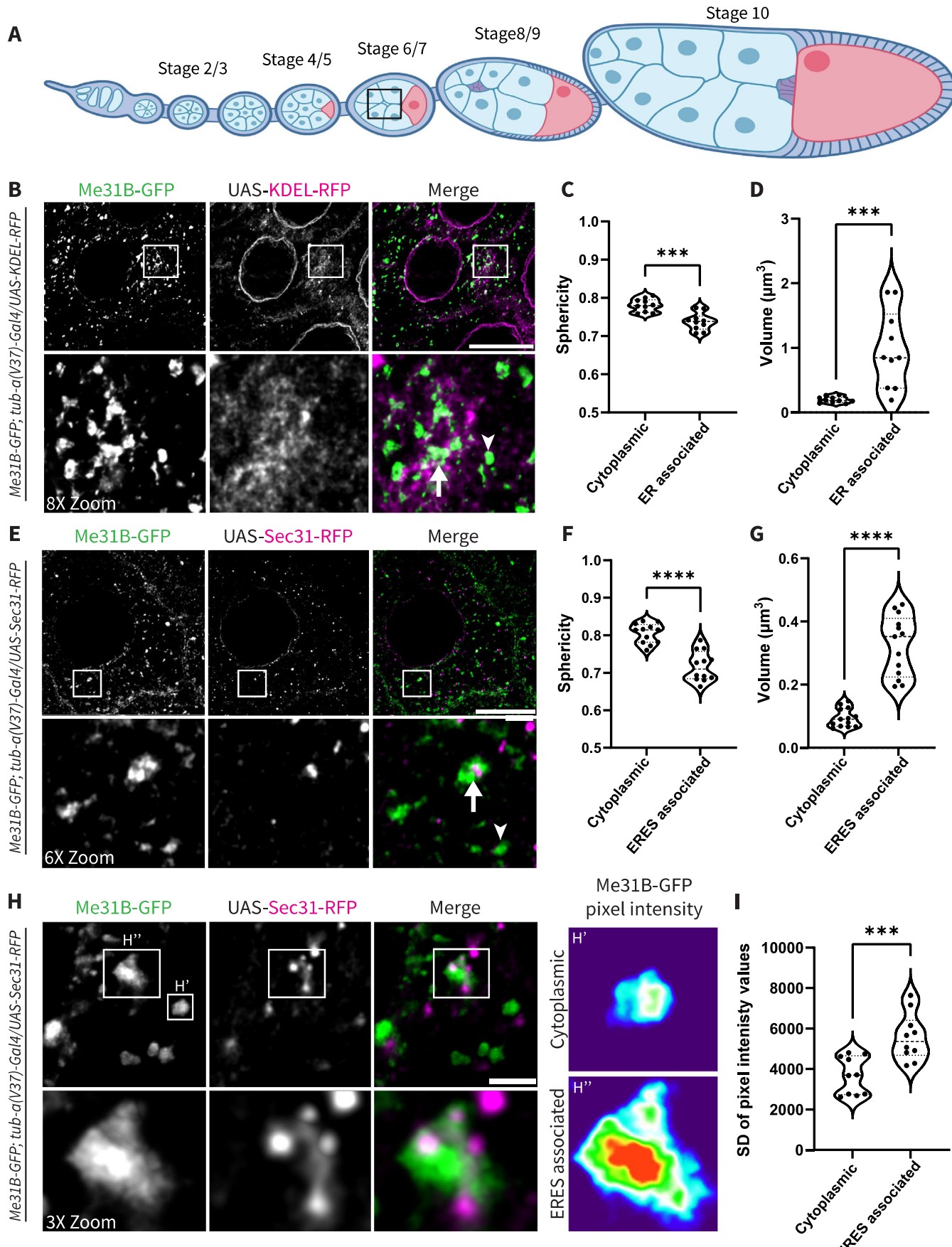

**Figure 1. P-bodies colocalized with ER exit sites are distinct from cytoplasmic P-bodies.**

(A) Schematic of a fly ovariole. Development stages progress from left to right. The oocyte (pink) and the nurse cells (light blue) are surrounded by somatic follicles (darker blue). The outlined ROI (black square) indicates the nurse cell region imaged for all zoomed image panels. (B) Endogenous Me31B-GFP colocalized with UAS-KDEL-RFP in nurse cell cytoplasm. ER-associated P-body (arrow) and cytoplasmic P-body (arrowhead) in Merge panel. Images are XY projections of 5 optical Z slices of 0.3 μm. Scale bars are 20 μm and 2.5 μm, respectively, in the zoomed inset. (C, D) Sphericity and volume measurements comparing cytoplasmic and ER-associated P-bodies (n = 10, biological replicates) P value = 0.0002 and 0.0006, respectively. (E) Endogenous Me31B-GFP colocalization with UAS-Sec31-RFP in nurse cell cytoplasm. ERES-associated P-body (arrow) and cytoplasmic P-body (arrowhead) in Merge panel. Images are XY projections of 5 optical Z slices of 0.3 μm. Scale bars are 20 μm and 3.3 μm, respectively, in the zoomed inset. (F, G) Sphericity and volume measurements of cytoplasmic and ERES-associated P-bodies (n = 10, biological replicates). P values < 0.0001. (H) Visualization Me31B-GFP with UAS-Sec31-RFP via STED. Images are XY projections of 5 optical Z slices of 0.22 μm. Scale bars are 2 μm and 670 nm, respectively, in the zoomed inset. (H′, H″) Intensity heat map of a cytoplasmic and ERES-associated P-body, respectively. (I) Standard deviation of pixel intensity values for cytoplasmic and ERES-associated P-bodies (n = 10, biological replicates). P value = 0.0008. All images were acquired at stage 7 of development. For all plots, each data point represents the average value of all P-bodies detected in an image. Significance was assessed using a Mann–Whitney statistical test. Error bars represent standard deviation. ****$P < 0.0001$. Source data are available online for this figure.

Given the possibility that the sphericity of condensates might be affected by the crowded, tubulovesicular nature of the ER, we compared the sphericity of ER-associated P-bodies and ERES-associated P-bodies. As all P-bodies in both datasets were colocalized with the ER membrane, we believe that any effects of crowding are eliminated and ERES-associated P-bodies are less spherical than ER-associated P-bodies, suggesting that they may be less liquid-like (Fig. EV1D).

As the liquid-like state of condensates has been shown to influence P-body function, we asked if ERES-associated P-bodies exhibit a different physical state than cytoplasmic P-bodies (Sankaranarayanan et al, 2021). To address this question, we employed STED super-resolution microscopy to visualize condensate 'texture' which provides insight into condensate physical state (Fig. 1H). Texture assesses the distribution of fluorophores within a cross section of a condensate, thus providing a readout of the internal structure of a P-body. 'Rougher' P-bodies indicate more solid/gel-like condensates as fluorescence signal is distributed in clumped structures, while 'smoother' P-bodies indicate more liquid-like condensates, with more evenly dispersed fluorescence signal (Shiina, 2019).

To quantify fluorescence distribution within a P-body, we utilized standard deviation (SD) of pixel intensity (Irgen-Gioro et al, 2022). This technique requires assessing the intensity of each pixel within the boundary of a P-body and obtaining a SD which indicates the range of pixel intensities within the condensate. Since 'rough' P-bodies contain fluorescence peaks and valleys, with high and low pixel intensities, respectively (Fig. 1H″), they have a larger SD of pixel intensity when compared to 'smooth' P-bodies which present pixels with more homogenic intensity values (Fig. 1H′). For each detected cytoplasmic and ERES-associated, a SD of pixel intensity value was calculated. Using this analysis, we found that ERES-associated P-bodies were ~34% 'rougher' than cytoplasmic P-bodies (Fig. 1I). This may indicate that these P-bodies are taking on gel-like condensate states, and as condensates were shown to 'harden' over time in vitro and in silica, it may also suggest that these condensates are more mature (Garaizar et al, 2022; Lin et al, 2015).

## ERES-associated P-bodies are dynamically distinct from cytoplasmic P-bodies

P-bodies display multiple patterns of dynamics; some stay stationary, others move within local vicinities, and some move along the microtubule network (Aizer and Shav-Tal, 2008). To attain a better understanding of how ERES-associated P-bodies

differ from cytoplasmic P-bodies, we performed live imaging to visualize Me31B-GFP with Sec31-RFP. Interestingly, ERES-associated P-bodies did not appear to move long distances and were often imaged throughout the entirety of the live imaging session (Fig. 2A), while cytoplasmic P-bodies were only imaged over a few time points, indicating that they move long distances (Fig. 2B). We next calculated track linearity and found that cytoplasmic P-bodies have ~46% more linear tracks compared to ERES-associated P-bodies (Fig. 2C). Concurrently, we found that cytoplasmic P-bodies moved 10% faster than ERES-associated P-bodies (Fig. 2D). Perhaps indicating that ERES-associated P-bodies are less liquid-like than cytoplasmic P-bodies.

To further investigate the differences in their dynamics, we employed fluorescence recovery after photobleaching (FRAP). We analyzed the colocalization of Me31B-GFP condensates with ERES in small areas of stage 7 egg chambers. After bleaching, we allowed for fluorescence recovery and recalculated their colocalization (Fig. EV2A). Interestingly, we observed that after photobleaching, there were 57% more cytoplasmic P-bodies in the field of view compared to ERES-associated P-bodies (Fig. EV2B). This suggests that cytoplasmic P-bodies migrated into the bleached area while ERES-associated P-bodies remained relatively stationary.

A recent study identified Me31B as part of the Egalitarian interactome, suggesting that at least some P-bodies move along microtubules via the Egalitarian/BicD/Dynein complex (Baker et al, 2021). To determine if the movement of cytoplasmic P-bodies, and not of ERES-associated P-bodies, relied on association with microtubules, we knocked down BicD using RNAi to prevent P-body attachment to microtubules and assessed Me31B-GFP dynamics by calculating track Mean Squared Displacement (MSD) (Fig. 2E,F). The knockdown's efficiency was confirmed via immunofluorescence (Fig. EV2C). Using this analysis, we found that cytoplasmic P-bodies traveled 2.5 times further than ERES-associated P-bodies within a 45 s time frame. Notably, in the $bicD^{RNAi}$ background, P-bodies displayed MSDs more comparable to ERES-associated P-bodies than cytoplasmic P-bodies, further suggesting that cytoplasmic P-bodies do utilize the microtubule network while ERES-associated P-bodies do not (Fig. 2F).

## Cytoplasmic P-bodies are removed from ERES-associated P-bodies via the microtubule network

Having established that ERES-associated P-bodies are larger and less dynamic than their cytoplasmic counterparts, we aimed to

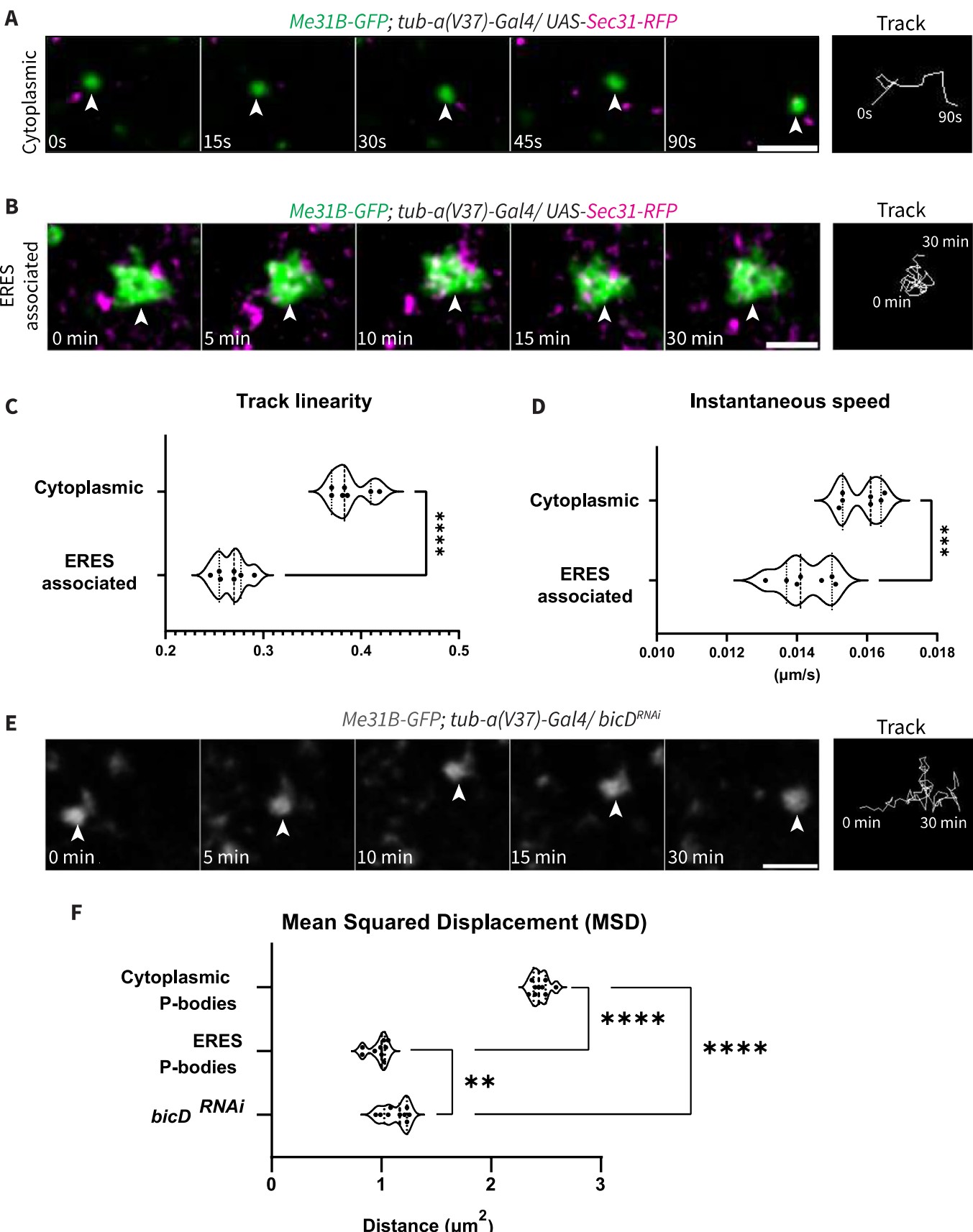

**A** *Me31B-GFP; tub-a(V37)-Gal4/ UAS-Sec31-RFP*

Cytoplasmic

Track

0s 15s 30s 45s 90s

0s 90s

**B** *Me31B-GFP; tub-a(V37)-Gal4/ UAS-Sec31-RFP*

ERES associated

Track

0 min 5 min 10 min 15 min 30 min

30 min

0 min

**C** Track linearity

Cytoplasmic

ERES associated

****

0.2 0.3 0.4 0.5

**D** Instantaneous speed

Cytoplasmic

ERES associated

***

0.010 0.012 0.014 0.016 0.018
(µm/s)

**E** *Me31B-GFP; tub-a(V37)-Gal4/ bicD^RNAi*

Track

0 min 5 min 10 min 15 min 30 min

0 min 30 min

**F** Mean Squared Displacement (MSD)

Cytoplasmic P-bodies

ERES P-bodies

*bicD* $^{RNAi}$

****

****

**

0 1 2 3
Distance (µm$^2$)

**Figure 2.  ERES-associated P-bodies are dynamically distinct from cytoplasmic P-bodies.**

(A) Tracking of cytoplasmic P-body (arrowhead). (B) Tracking of ERES-associated P-body (arrowhead). (C) Track linearity calculated for cytoplasmic and ERES-associated P-bodies ($n = 7$, biological replicates). $P$ value < 0.0001. (D) Instantaneous speed calculated for cytoplasmic and ERES-associated P-bodies. Each point represents the average speed at a single time point ($n = 7$, biological replicates). (E) Tracking of Me31B-GFP in *bicD*[RNAi] nurse cell (arrowhead). Images acquired every 15 s for 30 min. (F) Mean Squared Displacement of cytoplasmic P-bodies, ERES-associated P-bodies, and P-bodies in *bicD*[RNAi] nurse cells over a period of 45 s ($n = 9$, biological replicates). ERES-associated P-bodies vs. cytoplasmic P-bodies, $P < 0.0001$. ERES-associated P-bodies vs. *bicD*[RNAi] P-bodies, $P = 0.0083$. Cytoplasmic P-bodies vs. *bicD*[RNAi] P-bodies, $P < 0.0001$. All tracking was done in nurse cells of stage 7 egg chambers. All scale bars are 2 μm. For all plots, each point represents the average value of all P-bodies in an image. Significance was calculated using a Mann–Whitney statistical test. Error bars represent standard deviation. ****$P < 0.0001$. Source data are available online for this figure.

investigate their functional role. To accomplish this, we again utilized live imaging. Over a 30-min imaging period, we determined that ERES-associated P-bodies exhibited track durations of approximately 13 min, indicating that their interaction is not transient (Fig. EV2D). Notably, we observed no instances of fusion at these sites, while numerous instances of fission were recorded (Fig. 3A). Given that basic thermodynamics suggests that condensates should increase in size to minimize surface tension, the observed fission likely resulted from external forces acting on the ERES-associated P-bodies.

This led us to postulate that cytoplasmic P-bodies may be removed from ERES-associated P-bodies by the microtubule network. To investigate this, we visualized Me31B-GFP and Sec31-RFP in a *bicD*[RNAi] background (Fig. 3B). Microtubule expression was unaffected in the *bicD*[RNAi] egg chambers so alterations in cytoplasmic crowding or viscosity did not affect our results (Fig. EV2E). Interestingly, upon BicD knockdown, Me31B-GFP colocalization with ERES increased by 66% (Fig. 3C). Suggesting that, without connection to microtubules, the population of P-bodies shifts dramatically to ERES-associated P-bodies. As it is possible that cytoplasmic P-bodies require a connection to microtubules to form/be maintained, we assessed the volume of ERES-associated Me31B-GFP condensates in the *bicD*[RNAi] background. If cytoplasmic P-bodies were falling apart, ERES-associated P-body size should be unaltered by the knockdown. However, we observed that when cytoplasmic P-bodies were unable to move via microtubules, ERES-associated P-bodies became 55% larger than in *mCherry*[RNAi] control egg chambers (0.646 μm³ compared to 0.416 μm³) (Figs. 3D and EV2F). This suggests that they may be undergoing fewer fission events when P-bodies were unable to utilize the Egalitarian/BicD/Dynein complex.

To ensure that this was a result of P-bodies not associating with the microtubule network, we chose to visualize egg chambers where microtubules were destabilized with 10 μM colcemid for 30 min (Fig. EV2G). Notably, ERES-associated P-bodies again represented a larger percentage of the condensate population, 65% more than in the non-treated control (Fig. 3E,F), further suggesting the cytoplasmic population of P-bodies depends on the microtubule network. Taken together this data indicates that ER-associated P-bodies are a distinct class of P-body we are calling ERES P-bodies, which may function as hubs for cytoplasmic P-body formation.

## COPII vesicle proteins affect the organization of putative P-body protein condensates

As ERES P-bodies were quantitatively distinct from cytoplasmic P-bodies, we sought to better decipher the connection between

P-bodies and ERES components. ERES are composed of a suite of conserved proteins. Sar1, a cytoplasmic GTPase, is thought to govern the assembly of ERES (Van Der Verren and Zanetti, 2023). Upon interaction with the membrane-bound GEF, Sec12, the GTP-bound Sar1 inserts into the membrane and facilitates deformation. By inducing curvature, Sar1 insertion enhances the affinity of additional Sar1 for the site and thus propagates vesicle formation (Hanna et al, 2016). In *D. melanogaster*, Sec16, a large scaffolding protein, is recruited to ERES independently of Sar1 (Ivan et al, 2008). Once these two proteins are at the ER membrane, they assemble the inner COPII vesicle coat made up of Sec23 and Sec24 which subsequently recruits the outer coat of Sec13 and Sec31 (Fig. 4A) (Kurokawa and Nakano, 2019).

To probe the relationship between P-bodies and ERES, we separately analyzed three endogenously tagged P-body proteins: Me31B, Tral, and Cup, in the background of all publicly available COPII vesicle coat-protein RNAi lines: *sec23*[RNAi], *sec13*[RNAi], and *sec31*[RNAi] (Fig. 4B). The RNAi line efficiencies were validated with antibodies when available, or RT-qPCR (Fig. EV3A–C). Interestingly, in these knockdown backgrounds, all P-bodies appeared to be similarly affected, despite ERES-P-bodies constituting only ~20% of the total population. This further bolsters the idea that ERES serve as hubs for P-body formation. Given this observation, we decided to quantify all nurse cell P-bodies to acquire a holistic view of the effects of COPII knockdowns on the global P-body population.

We first knocked down the inner-coat protein, Sec23, which also functions as the GTPase-activating protein (GAP) responsible for terminating COPII budding (Yoshihisa et al, 1993). In *sec23*[RNAi] egg chambers, Me31B-GFP condensates were 17% more spherical (Fig. 4C). These condensates were also 86% larger than in the control background (Fig. 4D). Similarly, Tral condensates were 8% more spherical and 59% larger compared to condensates in the control (Fig. 4E,F). This indicates that P-body maintenance is severely compromised, and overall P-body organization is aberrant in the *sec23*[RNAi] background. Notably, Cup condensates were unaffected, suggesting that proper COPII vesicle formation is not as important for Cup organization within P-bodies as it is for Me31B and Tral (Fig. 4G,H).

Next, we investigated the outer-coat proteins: Sec13 and Sec31. The P-bodies in these knockdowns again exhibited altered organization. Me31B-GFP condensates were on average 18% more spherical in the *sec13*[RNAi] egg chambers and 19% more spherical in *sec31*[RNAi\] egg chambers (Fig. 4C). Condensates were also 58% and 21% larger, respectively, in the knockdown egg chambers compared to control (Fig. 4D). Interestingly, Tral-RFP condensates did not display sphericities that differed significantly from the control but were on average 33% and 122% larger, respectively, when compared

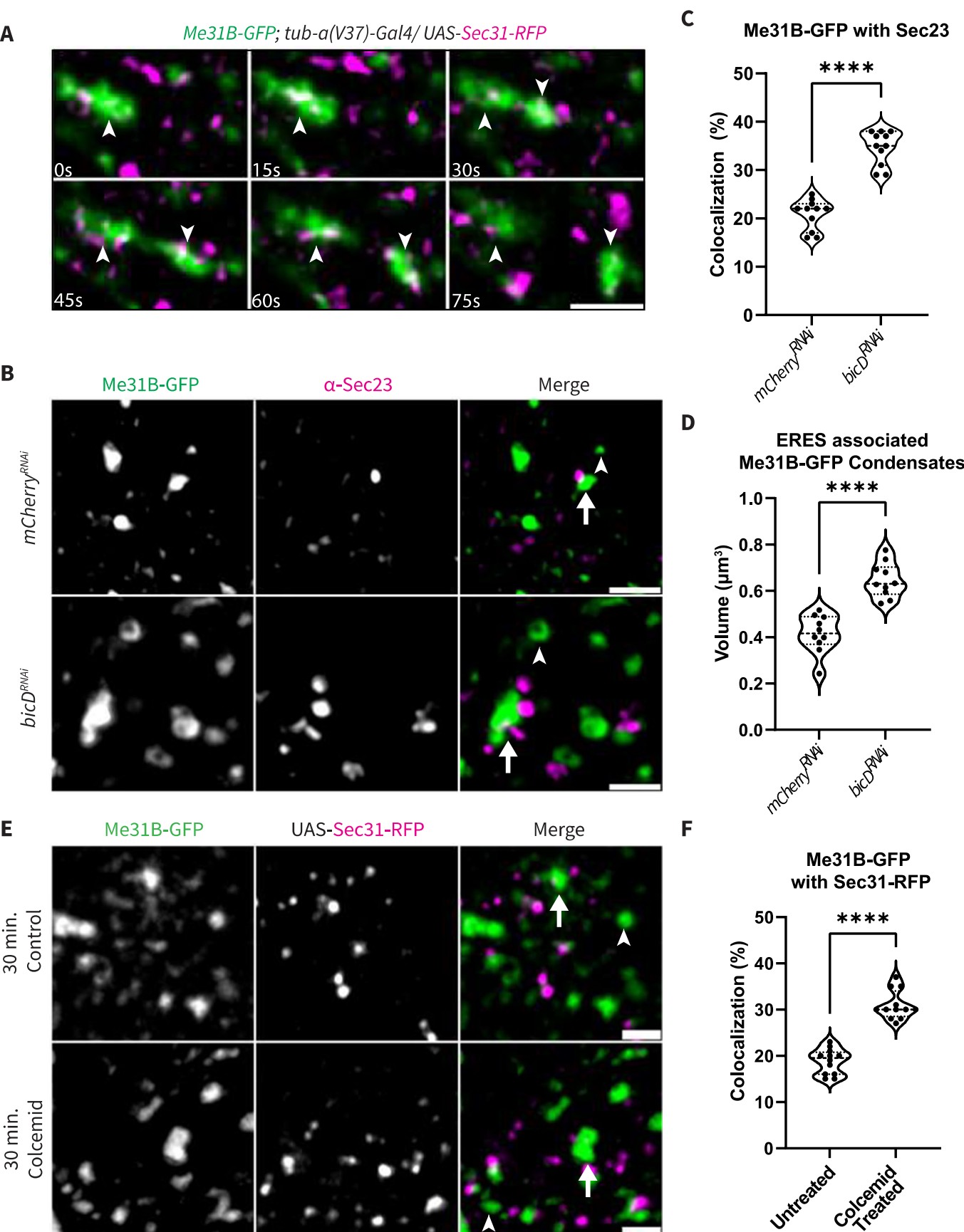

**Figure 3. Cytoplasmic P-bodies are removed from ERES-associated P-bodies by the microtubule network.**

(A) P-body fission event at ERES. Arrowheads represent separate condensates. (B) Me31B-GFP and Sec23 co-visualized with immunofluorescence in *mCherry*^RNAi^ and *bicD*^RNAi^ nurse cells. ERES-associated P-body (arrow) and cytoplasmic P-body (arrowhead) in Merge panel. XY projections of 5 optical Z slices of 0.3 μm. (C) Colocalization analysis of Me31B-GFP with Sec23, calculated with "Surface Detection" on Imaris software ($n = 10$, biological replicates). $P < 0.0001$. (D) Volume of Me31B-GFP condensates colocalized with Sec23 in *mCherry*^RNAi^ control and *bicD*^RNAi^ nurse cells ($n = 11$, biological replicates). $P < 0.0001$. (E) Me31B-GFP with Sec31-RFP incubated in Schneider's media (control) or 10 μM colcemid for 30 min. ERES-associated P-body (arrow) and cytoplasmic P-body (arrowhead) in Merge panel. XY projections of 5 optical Z slices of 0.3 μm. (F) Colocalization analysis of Me31B-GFP with Sec31-RFP ($n = 10$, biological replicates). $P < 0.0001$. All images are from stage 7 egg chambers. All scale bars are 2 μm. For all plots, each point represents the average value of all P-bodies in an image. Significance was calculated using a Mann–Whitney statistical test. Error bars represent standard deviation. ****$P < 0.0001$. Source data are available online for this figure.

to control egg chambers (Fig. 4E,F). Again, Cup was unaffected by alteration in COPII vesicle composition (Fig. 4G,H).

Together this data suggests that proper COPII organization is imperative for Me31B and Tral condensate organization but inconsequential for Cup organization. This is of interest as all three proteins are known to form a conserved ribonucleoprotein (RNP) (Nakamura et al, 2004; Tritschler et al, 2008). This indicates that COPII vesicles may play a crucial role in dictating overall P-body organization. Given this, we aimed to investigate the impact of attenuating overall COPII formation on P-body organization.

Sec16 is an important protein for dictating ERES localization and COPII vesicle formation initiation (Sprangers and Rabouille, 2015). The only Sec16 knockdown line available provides a ~60% knockdown efficiency (Fig. EV3D). This allowed us to quantify P-bodies in a background of compromised, but not ablated, COPII vesicle formation (Fig. 4I). Notably, in *sec16*^RNAi^ egg chambers, Me31B-GFP condensates were 28% more spherical and 37% smaller compared to condensates in *mCherry*^RNAi^ egg chambers, indicating that Me31B-GFP condensates are either unable to fully form, or maintain, proper condensate state (Fig. 4C,D) (Bayer et al, 2023; Sankaranarayanan et al, 2021). Similarly, when we analyzed Tral-RFP condensates in this background, they were on average 7% more spherical and 51% smaller compared to control egg chambers (Fig. 4E,F). Again, Cup condensates were unaffected (Fig. 4G,H). These results are strikingly different from the COPII vesicle knockdowns. While the coat protein knockdowns seemed to disrupt organization, the Sec16 knockdown seemed to result in improper P-body formation or compromised P-body maintenance as Me31B and Tral condensates were significantly smaller in volume and more spherical suggesting that they are more liquid-like and less structured than proper P-bodies.

As knocking down ERES proteins may have numerous pleiotropic effects that could influence P-body organization, we chose to assess the most plausible candidates. First, we considered that these knockdowns could lead to alterations in P-body protein levels as the cause of the change in P-body organization. However, Western blot analysis showed that overall protein levels were unaffected (Fig. 4J).

Given the previously demonstrated influence of the ER on microtubule arrangement, which in turn may affect P-body formation, we postulated that the disruption in condensate organization may arise from aberrant microtubule arrangement in the ERES knockdown backgrounds (Sweet et al, 2007; Tikhomirova et al, 2022). To assess this, we visualized the integrity of microtubules, F-actin, and nuclear membranes in *sec16*^RNAi^, *sec23*^RNAi^, *sec13*^RNAi^, and *sec31*^RNAi^ backgrounds and found that they were unaffected, suggesting that the condensate alterations were

not a result of cellular breakdown (Fig. EV3F). We also considered that the phenotypes may result from gross alterations in the ER. However, visualization of KDEL-RFP indicated that the overall ER membrane was unaffected in the knockdown backgrounds (Fig. EV3G).

Finally, we considered that the presence of large condensates might be a consequence of stress. This is plausible as stress granules contain many of the same proteins as P-bodies (Ivanov et al, 2019). To assess this, we visualized the distribution of Rasputin (Rin), a stress granule marker, in each of the COPII protein knockdown backgrounds. Notably, we did not observe any Rin-GFP condensates, indicating the absence of stress granule formation (Fig. EV3H). Taken together, these findings lead us to believe that condensate mis-organization is likely a consequence of improper COPII assembly.

## In the absence of ER exit sites, P-body integrity is compromised

Since our data suggests that ERES may play a critical role in P-body formation, we aimed to visualize P-bodies in a background without any ERES formation initiation. This was achieved by knocking down Sar1, verified with RT-qPCR (Fig. EV4A). This RNAi line should knockdown COPII initiation by preventing ERES formation similar to Sec16, however, as it is more penetrant, it provides a complete ablation of ERES formation (Fig. EV4B).

Since egg chambers subjected to Sar1 knockdowns do not fully develop, we opted to initially induce the knockdown using a weaker Gal4 line (*otu*-Gal4). This Gal4 permits mild expression of Sar1, resulting in a less severe phenotype. By visualizing Me31B, Tral, and Cup in this background, we determined that only a limited number of P-bodies were present and P-body proteins were more diffuse (Fig. 5A, with B, C representing distribution of pixel values from ROI-red box of the Merge panels). This observation was surprising since a study in yeast found that in the absence of Sar1, there was an increase in P-body number (Kilchert et al, 2010). This difference may serve as evidence of the various pathways of P-body formation that are implemented across species and cell types.

To determine if the decrease in Sar1 was responsible for the limited number of P-bodies, we visualized putative P-body proteins: Me31B, Cup, and Bruno 1, in the background of a strong Gal4 line (*a-tub*-Gal4) that induced a more substantial knockdown of Sar1 (Fig. 5D). We chose to visualize stage 7 egg chambers, as Sar1 knockdowns only developed until stage 8 and we did not want to assess cells that could potentially be undergoing cell death. In this background, we observed that each protein was completely diffuse in the nurse cell cytoplasm (Fig. 5E,F representing

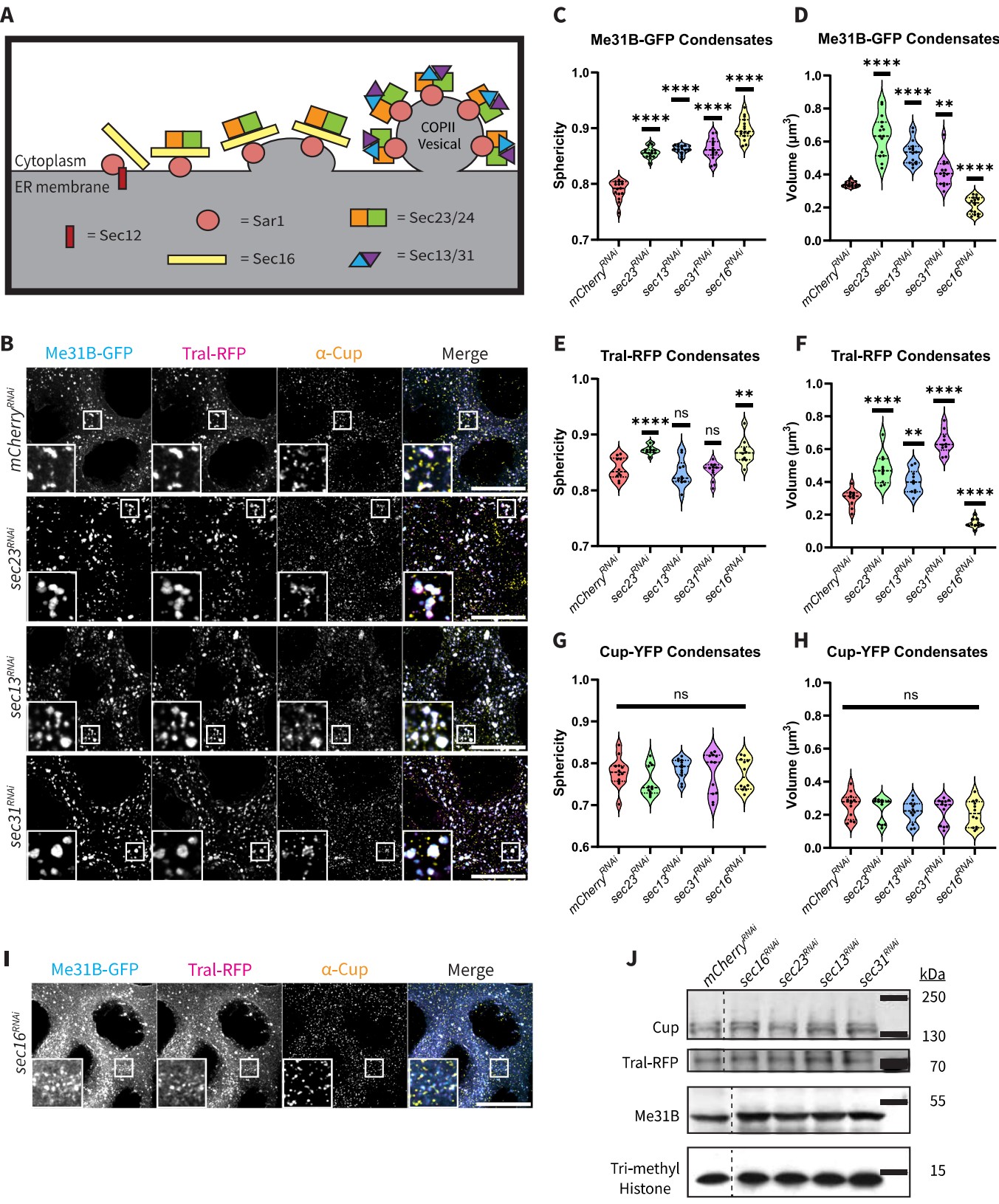

**Figure 4. COPII vesicle proteins affect the organization of putative P-body protein condensates.**

(A) Schematic of protein composition during ERES formation. (B) Co-visualization of Me31B-GFP, Tral-RFP, and Cup in nurse cells with systematic knockdown of individual COPII coat components. XY projections of 5 optical Z slices of 0.3 μm. Scale bars are 20 μm. (C) Sphericity measurements of Me31B-GFP condensates ($n = 15$, biological replicates). For all, $P < 0.0001$. (D) Volume measurements of Me31B-GFP condensates ($n = 15$, biological replicates). For $sec23^{RNAi}$, $sec13^{RNAi}$, and $sec16^{RNAi}$, $P < 0.0001$. For $sec31^{RNAi}$, $P = 0.0024$. (E) Sphericity measurements of Tral-RFP condensates ($n = 10$, biological replicates). For $sec23^{RNAi}$, $P < 0.0001$. For $sec13^{RNAi}$ and $sec31^{RNAi}$, $P = $ n.s. For $sec16^{RNAi}$, $P = 0.0056$. (F) Volume measurements of Tral-RFP condensates ($n = 10$, biological replicates). For $sec23^{RNAi}$, $sec31^{RNAi}$, and $sec16^{RNAi}$, $P < 0.0001$. For $sec13^{RNAi}$, $P = 0.0016$. (G, H) Sphericity and volume measurements for Cup-YFP condensates ($n = 12$, biological replicates). For all, $P = $ n.s. (I) Co-visualization of Me31B-GFP, Tral-RFP, and Cup in Sec16 knockdown nurse cells. XY projections of 5 optical Z slices of 0.3 μm. Scale bars are 20 μm. (J) Western blot of Cup, Tral-RFP, and Me31B. Tri-methyl-Histone — loading control in respective lysates of ERES knockdown egg chambers. All lysates were analyzed on a single gel, with the dashed lines indicating regions where the blot was spliced for clarity. All quantifications (C–H) were performed on egg chambers expressing only one fluorescently tagged protein at a time. All images represent nurse cells of stage 7 egg chambers. Significance was calculated using a Mann–Whitney statistical test. Error bars represent standard deviation. ****$P < 0.0001$. Source data are available online for this figure.

distribution of pixel values from ROI-red box of the Merge panels of Fig. 5D). Due to constraints of the *D. melanogaster* genome, we were unable to visualize Tral-RFP in this background. Notably, the stronger Gal4 line resulted in the complete diffusion of P-body components, while the weaker Gal4 line led to only an attenuation of P-body number. This finding is noteworthy, as it establishes a robust, causative link between ERES formation and P-body formation, which has not been previously demonstrated.

As the Sar1 knockdown induced cell death at stage 8, we were concerned that the altered P-body distribution might be a consequence of the early-stages of autophagy. To investigate this possibility, we employed LysoTracker to monitor acidic organelles indicative of autophagy but observed no signs of autophagic activity during the stages analyzed (Fig. EV4C). To further rule out the role of autophagy in the diffuse P-body phenotype, we examined P-bodies in egg chambers from flies subjected to 24 h of starvation to induce autophagy (Barth et al, 2011). Notably, in this context, P-bodies appeared larger and more punctate, in stark contrast to the diffuse distribution observed in the Sar1 knockdown (Fig. EV4C). Collectively, these results indicate that autophagy is not responsible for the diffusion of P-body proteins.

To determine if the reduced P-body population was a consequence of lower P-body protein concentrations in the $sar1^{RNAi}$ egg chambers, we conducted a Western blot analysis (Fig. 5G). Interestingly, the overall level of P-body proteins did not significantly change, showing that the lack of P-bodies in the Sar1 knockdown is not attributable to alterations in P-body protein levels, but rather to changes in their distributions.

As many maternal mRNAs are recruited into P-bodies, we further assessed whether mRNA distribution was also affected by the ERES knockdown (Weil et al, 2012). First, we aimed to establish the presence of putative P-body mRNAs within ERES P-bodies, as that may imply a reliance on ERES for mRNA granule formation. To investigate this, we co-visualized Me31B, Sec31, and several maternal mRNAs. Significantly, our findings revealed the colocalization of *oskar*, *bicoid*, and *nanos* mRNAs with ERES P-bodies (Figs. 5H and EV4D,E). We next visualized *ATP synthase* mRNA, a housekeeping transcript that is not enriched in P-bodies, and *oskar* mRNA, a known mRNA stored in P-bodies, in $sar1^{RNAi}$ egg chambers (Bayer et al, 2023; Kato and Nakamura, 2012). Notably, we found that the distribution of *ATP synthase* mRNA was unaffected, while *oskar* mRNA no longer formed large puncta in the nurse cell cytoplasm and was mislocalized to the anterior of the oocyte (Fig. 5I,J). This demonstrates that ERES are essential for the

assembly of *oskar* mRNA into granules and, consequently, for its proper localization within the oocyte.

Given the shared characteristics between P-bodies and stress granules (i.e. composed of similar proteins and mRNAs, form via LLPS, and are expressed in the cytoplasm), we wanted to ascertain if disruption of ERES similarly impacted stress granule formation (Corbet and Parker, 2019). To investigate this, we subjected females to short-term starvation to induce stress, and visualized Me31B, *ATP synthase* mRNA and *oskar* mRNA in control and $sar1^{RNAi}$ egg chambers. Surprisingly, we observed that Me31B and *oskar* were able to assemble into stress granules in these egg chambers, suggesting that ERES are exclusively crucial for P-body formation and do not compromise the formation of other LLPS granules (Fig. 5K).

## Sar1 knockdown leads to attenuation of P-body function

As P-bodies host multiple mRNA processes, we wanted to ascertain if these pathways were compromised in the absence of ERES. We focused our studies on *oskar* mRNA as it forms a well-studied mRNP and is known to localize into P-bodies (Bayer et al, 2024; Nakamura et al, 2001). First, we investigated ERES's effect on translational repression by expressing endogenous Oskar-GFP in a $sar1^{RNAi}$ background. *oskar* mRNA is transcribed in the nurse cells throughout oogenesis and localizes posteriorly in the oocyte where Oskar protein is expressed during mid and late oogenesis (Riechmann and Ephrussi, 2001). Interestingly, we found that, in $sar1^{RNAi}$ egg chambers, Oskar was prematurely expressed in the early oocyte (Fig. 6A).

Next, we investigated how the absence of Sar1 affected P-body's role in maintaining *oskar* mRNA stability. Using RT-qPCR, we found an average 51% decrease in *oskar* mRNA levels (Fig. 6B). To determine if this decrease was due to diminished RNA transcription or stability, we assessed the volume of active transcription sites in $sar1^{RNAi}$ egg chambers and determined that transcription was not significantly altered (Fig. 6C). To assess transcript stability, we used two sets of smFISH probes. One set detected the 5′ UTR of the *oskar* transcript and the other detected the 3′ UTR (Fig. 6D). Through imaging differentially labeled *oskar* 5′ UTR and 3′ UTRs, followed by colocalization analysis, we observed that ~40% more *oskar* transcripts were degraded in the $sar1^{RNAi}$ egg chambers compared to control. This suggests that the Sar1 knockdown resulted in reduced transcript stability in the cytoplasm (Fig. 6E).

Using this method, we also established that *oskar* mRNA was degraded in both directions, but primarily underwent degradation

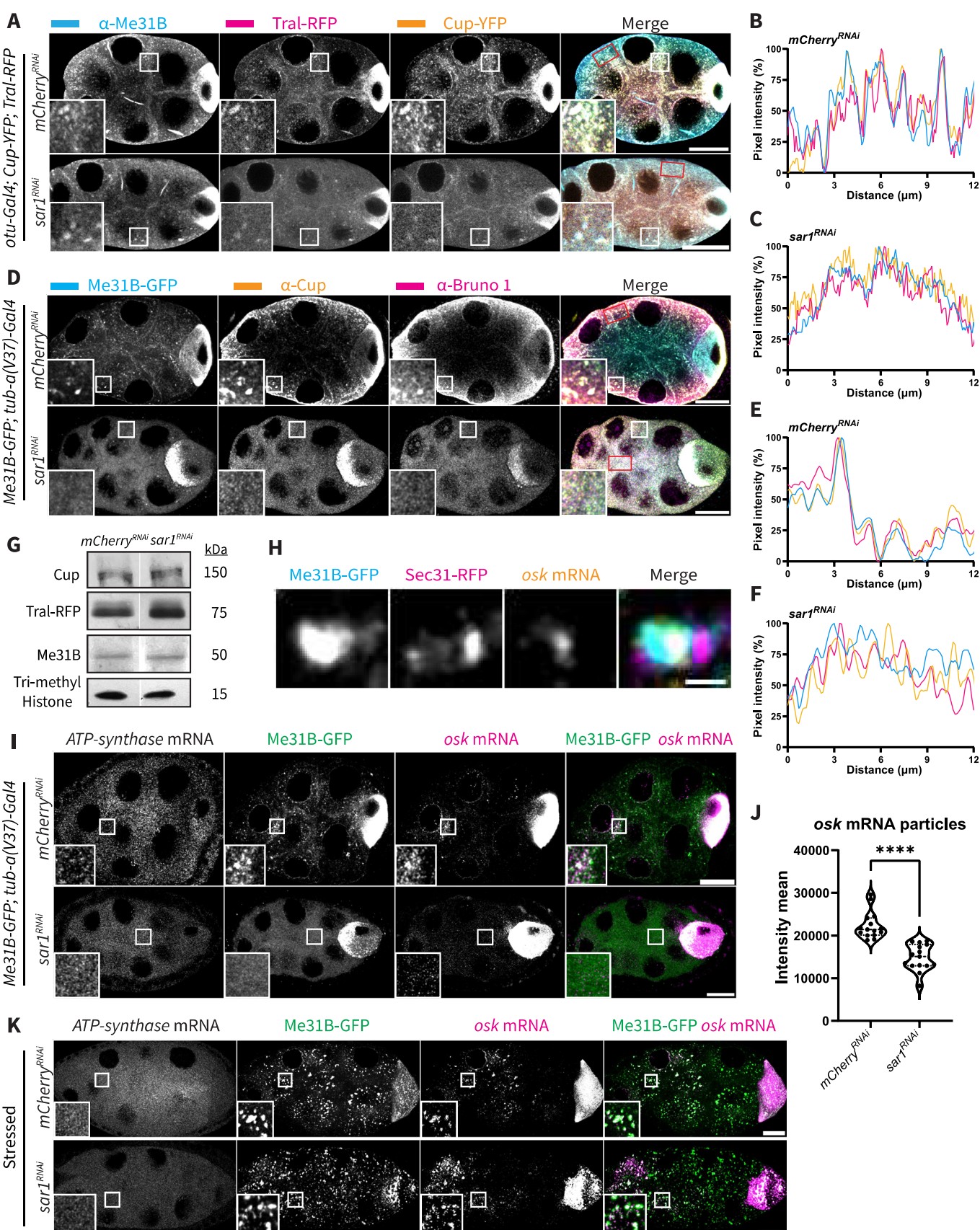

**Figure 5. In the absence of ER exit sites, P-body integrity is compromised.**

(A) Co-visualization of Me31B, Tral-RFP, and Cup-YFP in *mCherry^RNAi* and *sar1^RNAi* egg chambers. (B, C) Line plots of pixel intensity in *mCherry^RNAi* and *sar1^RNAi* egg chambers, respectively, driven by *otu-GAL4*. Each plot represents a 100 pixel wide 12 μm ROI (red box in Merge panels). (D) Me31B-GFP co-visualized with Cup and Bruno 1 in *mCherry^RNAi* and *sar1^RNAi* egg chambers. Scale bars are 20 μm. (E, F) Line plots of pixel intensity in *mCherry^RNAi* and *sar1^RNAi* egg chambers, respectively, driven by α-*tub-GAL4*. Each plot represents a 100 pixel wide 12 μm ROI (red box in Merge panel). (G) Western blot analysis of Cup, Tral-RFP, and Me31B (Tri-methyl-Histone — loading control). (H) Me31B-GFP condensate colocalized with Sec31-RFP and *oskar* mRNA visualized with smFISH probes. Scale bar is 1 μm. (I) Me31B-GFP, *ATP synthase* mRNA, and *oskar* mRNA visualized with respective smFISH probes in *mCherry^RNAi* and *sar1^RNAi* egg chambers. (J) Intensity measurements of *oskar* mRNA particles in *mCherry^RNAi* and *sar1^RNAi* egg chambers ($n = 15$, biological replicates). $P < 0.0001$. (K) Me31B, *ATP synthase* mRNA and *oskar* mRNA visualized under nutritional stress in *mCherry^RNAi* and *sar1^RNAi* backgrounds. All images are XY projections of 5 optical Z slices of 0.3 μm. Scale bars are 20 μm. Significance was calculated using a Mann–Whitney statistical test. Error bars represent standard deviation. ****$P < 0.0001$. Source data are available online for this figure.

via the 3′ to 5′ decay pathway (Fig. 6F,G). We performed this analysis for each of the COPII knockdowns and found a similar decrease in transcript stability, although not as pronounced as in Sar1 knockdown egg chambers (Fig. EV5A–C). This suggests that aberrations in P-body morphology affect overall P-body function.

Given the established roles of P-body proteins Me31B, Cup, and Tral in translational repression and transcript stability, we asked whether compromised binding of any of these proteins to the transcript may be responsible for the observed phenotypes (Götze et al, 2017; Nakamura et al, 2004). To address this possibility, we used STED super-resolution imaging to visualize Me31B-GFP, a key P-body protein in part responsible for *oskar* mRNA translational repression (Nakamura et al, 2001), with *oskar* mRNA in *sar1^RNAi* egg chambers. Notably, we found that the colocalization between *oskar* transcripts and Me31B-GFP was maintained (Fig. 6H,I). Cup similarly maintained association with *oskar* transcripts in *sar1^RNAi* egg chambers (Fig. EV5D,E). Interestingly, in both instances, *oskar* puncta are spread out more uniformly throughout the cytoplasm, similar to the P-body proteins, in the Sar1 knockdown compared to the knockdown egg chambers. Overall, this indicates that it is the compromised P-body formation that contributes to an ease in translational repression and stability, rather than a breakdown at the mRNP level. These findings strongly suggest that proper P-body formation/organization plays a significant role in mediating translational repression efficiency and transcript stability.

## Discussion

Biological condensates are emerging as key regulators of cellular processes (Banani et al, 2017; Lyon et al, 2021). While our understanding of their functions is expanding, a comprehensive picture of how their formation is governed in complex tissue has yet to emerge. Previous studies have demonstrated the regulation of ERES localization in *D. melanogaster* by Tral (Wilhelm et al, 2005). Here, we show the reciprocal regulation: ERES are necessary for proper P-body formation/integrity during *D. melanogaster* oogenesis. We propose a model where ERES contributes to the formation/organization of larger, gel-like P-bodies that serve as nucleating hubs from which smaller, liquid-like P-bodies are removed by the cytoskeletal network and become more mobile. Further biochemical studies are necessary to demonstrate that the link described in this proposed model is direct, but as even an indirect connection would be of interest our findings justify further study.

### ERES P-bodies are a discrete class of P-bodies

Previous work has depicted P-bodies as a heterogeneous group of condensates, encompassing both liquid-like, mobile condensates, as well as solid-like, stationary condensates (Aizer et al, 2008; Ditlev et al, 2018). Here, we demonstrate that this diverse population may represent different types of P-bodies with varying characteristics, potentially indicative of different functional roles. We propose that ERES P-bodies constitute one of these distinct classes and function as hubs of P-body formation. Interestingly, ERES P-bodies were larger than cytoplasmic P-bodies, and while we visualized many fission events, we found no instances where we could detect fusion of P-bodies. This suggests that P-bodies may be forming at these sites possibly from individual RNPs or proteins, and not from the fusion of large, mature condensates.

Notably, the ER serves as a nucleating site for several LLPS-driven bodies. TIS granules, membrane-less bodies that house a specific subset of mRNAs, form in contact with the ER membrane (Ma and Mayr, 2018). Similarly, the assembly of Whi3 condensates in fungi is regulated, at least in part, by ER contact sites (Snead et al, 2022). Another type of LLPS condensate called Sec bodies also form at the ER under stress and are composed of ERES proteins (Zacharogianni et al, 2014). Since the presence of LLPS initiation at the ER is well established, we believe that the connection between P-body formation and ERES is a natural one. This idea is particularly compelling as ERES are involved in regulating long-distance protein localization, while P-bodies potentially work to regulate long-distance mRNA localization, making their functions somewhat parallel. The reciprocal regulation between P-bodies and ERES may serve a teleological function in establishing polarity over long distances.

### COPII vesicle components dictate P-body organization

P-bodies encompass a diverse array of proteins and transcripts. Me31B, Tral, and Cup are canonical P-body proteins that interact with each other and are interchangeably used as P-body markers. However, our findings reveal that these proteins are not uniformly affected by COPII knockdowns. While Me31B and Tral condensates exhibited similar dependence on proper COPII assembly, Cup remained unaffected. This underscores the necessity for a comprehensive understanding of how each of the proteins present in P-bodies contributes to overall condensate state and function. There are several explanations as to why these proteins are differentially affected. Perhaps they require different thresholds of ERES function to form condensates. Alternatively, it is possible that ERES only directly affects one of the P-body proteins, and changes

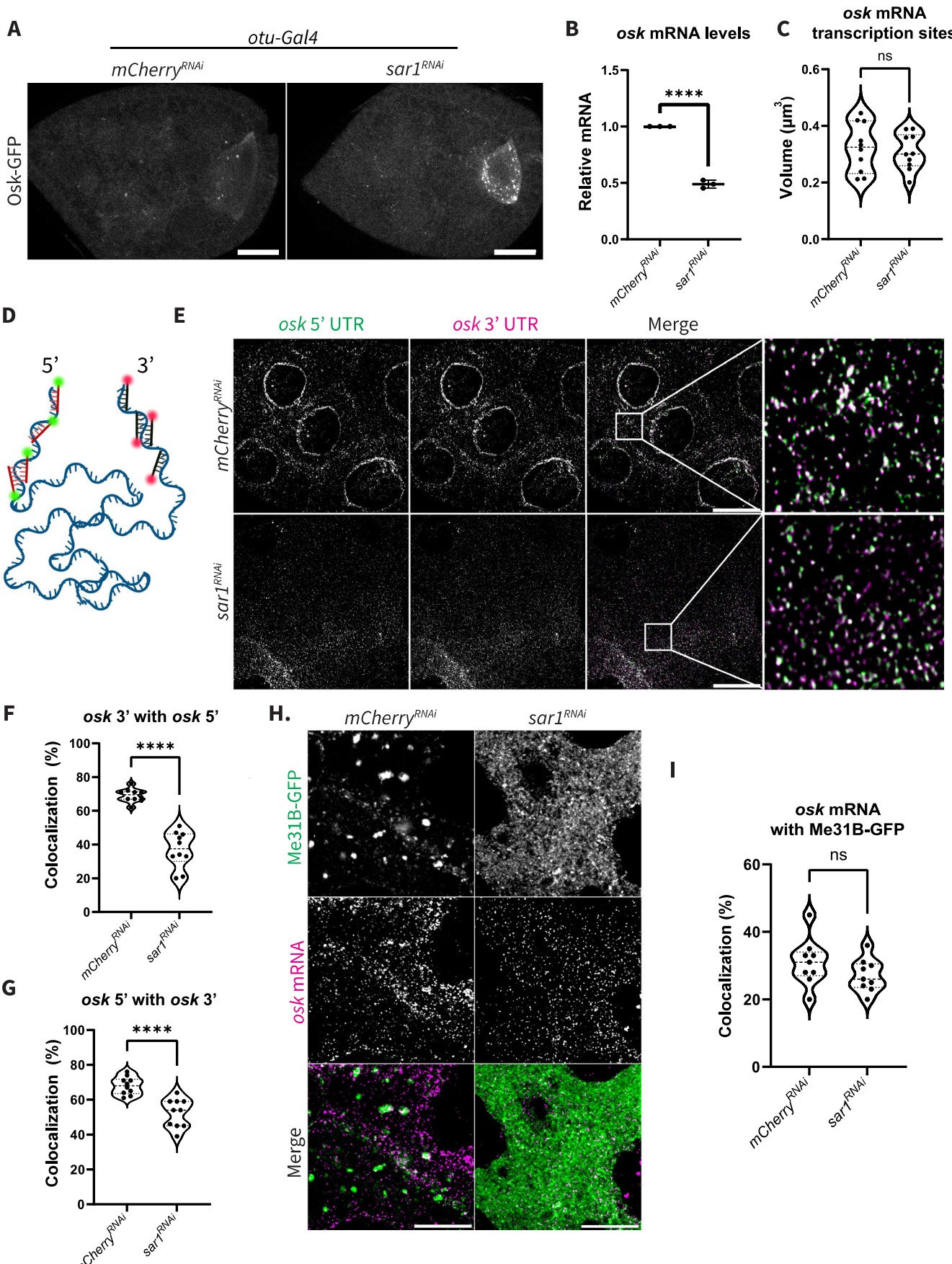

◀ **Figure 6.   Sar1 knockdown leads to attenuation of P-body function.**

(A) Oskar-GFP visualized in *mCherry*[RNAi] and *sar1*[RNAi] egg chambers. XY projections of 5Z optical slices of 0.3 μm. Scale bars are 20 μm. (B) *oskar* mRNA levels calculated with RT-qPCR in *mCherry*[RNAi] and *sar1*[RNAi] backgrounds. Significance calculated with a t-test. (n = 3, biological replicates). P < 0.0001. (C) Volume calculation of active transcription sites in *mCherry*[RNAi] and *sar1*[RNAi] egg chambers (n = 10, biological replicates). P = n.s. (D) Diagram of 5′ UTR and 3′UTR *oskar* smFISH probes. (E) Co-visualization of *oskar* 5′ UTR and *oskar* 3′ UTR regions in *mCherry*[RNAi] and *sar1*[RNAi] egg chambers. XY projections of 5Z optical slices of 0.3 μm. Scale bars are 20 μm. (F, G) Colocalization analysis of *oskar* 3′ UTR with *oskar* 5′ UTR (n = 10, biological replicates) and *oskar* 5′ UTR with *oskar* 3′ UTR, respectively. For both, P < 0.0001. (H) Co-visualization of Me31B-GFP and *oskar* mRNA via STED in *mCherry*[RNAi] and *sar1*[RNAi] nurse cells. XY projections of 3Z optical slices of 0.22 μm. Scale bars are 10 μm. (I) Colocalization analysis of *oskar* mRNA with Me31B-GFP (n = 9, biological replicates). P < 0.0001. All images were from nurse cells of stage 7 egg chambers. Significance calculated with Mann–Whitney statistical test. Error bars represent standard deviation. ****P < 0.0001. Source data are available online for this figure.

in this condensate influences the organization of the other. Future in vitro experiments could elucidate the direct influence of COPII assembly on each protein.

## P-bodies are necessary for optimal *oskar* mRNA regulation

Our findings suggest that P-bodies play a role in increasing the efficiency of translational repression and maintaining transcript stability. In the absence of ERES, P-body formation was compromised, and we observed increased *oskar* transcript degradation and premature translation. While there was a decrease in the efficiency of both pathways, translational repression and transcript stability were still mostly maintained in the absence of detectable P-bodies. Interestingly, we only observed premature expression of Oskar protein in the oocyte. We favor a model where translational repression can be sufficiently maintained in the nurse cells within the *oskar* mRNP, although this is not sufficient for fully maintaining *oskar* mRNA stability without recruitment into P-bodies. In the oocyte, however, where *oskar* mRNA is highly enriched, the extra layer of protection in P-bodies from the translational machinery is needed to maintain adequate translational repression. We are aware of the limitations of our imaging systems; therefore, we do not overlook the possibility that there is a threshold of protein expression below which we might not be able to detect. Overall, this data suggests that P-bodies are essential for optimal efficiency but are not indispensable to translational repression and RNA stability, which is in agreement with prior studies (Blake et al, 2024; Eulalio et al, 2007b).

## ERES communicate with P-bodies

The secretory pathway plays a vital role in relaying external signals into the internal environment of the cell. An RNAi screen revealed that at least 122 kinases and phosphatases affect ER export. Among these are members of canonical EGF signaling pathways. The same study found that an increase in mitogen activated protein kinase (MAPK) pathways led to phosphorylation of Sec16 by ERK which increased ERES numbers (Farhan et al, 2010). Since our data suggests that ERES either directly or indirectly influences P-body formation/maintenance and function, there may be an interesting connection between cancer and the interface between the ER and membrane-less organelles which warrants further investigation.

Here, we propose a paradigm where ERES serve as connection points between a membrane-bound organelle and membrane-less P-bodies. What we find most compelling about this model is that it

allows for positive feedback between P-bodies and ERES, offering an explanation for the abundance of both structures in some cell types compared to others. However, much remains to be understood about the relationship between these sites and biological condensates. Conducting studies in various tissues concurrently will be essential for gaining a comprehensive view of the far-reaching implications of this regulatory mechanism.

## Methods

**Reagents and tools table**

| Reagent/ Resource | Reference or Source | Identifier or Catalog Number |
|---|---|---|
| **Experimental models** | | |
| *UAS-mCherry*[RNAi] | Bloomington Drosophila Stock Center | BL #35785 |
| *Me31B-GFP* | Bloomington Drosophila Stock Center | BL #52530 |
| *UAS-KDEL-RFP* | Bloomington Drosophila Stock Center | BL #30910 |
| *UAS-Sec31-RFP* | Bloomington Drosophila Stock Center | BL #86532 |
| *tub-α(V37)-Gal4* | Bloomington Drosophila Stock Center | BL #7063 |
| *UAS-bicD*[RNAi] | Bloomington Drosophila Stock Center | BL #42929 |
| *UAS-sec16*[RNAi] | Bloomington Drosophila Stock Center | BL #53917 |
| *UAS-sec23*[RNAi] | Bloomington Drosophila Stock Center | BL #32365 |
| *UAS-sec13*[RNAi] | Bloomington Drosophila Stock Center | BL #32468 |
| *UAS-sec31*[RNAi] | Bloomington Drosophila Stock Center | BL #32878 |
| *UAS-sar1*[RNAi] | Bloomington Drosophila Stock Center | BL #32364 |
| *otu -Gal4* | Bloomington Drosophila Stock Center | BL # 58424 |
| *Cup-YFP* | Kyoto Drosophila Stock Center | DGRC 115-161 |
| *Rin-GFP* | Vienna Drosophila Resource Center | VDCR #318907 |
| *Oskar-GFP* | Gift from Dr. G. Gonsalves (Augusta University) | NA |
| *Tral-RFP* | Gift from Dr. St. Johnston (Gurdon Institute at the University of Cambridge) | NA |
| **Antibodies** | | |
| Mouse α-BicD | Deposited to the DSHB (Developmental Studies Hybridoma Bank) by Steward, R | 1B11 |
| Mouse α-Cup | Gift from Dr. A. Nakamura (Institute of Molecular Embryology and Genetics, Kumamoto University) | NA |

| Reagent/ Resource | Reference or Source | Identifier or Catalog Number |
|---|---|---|
| Mouse α-Me31B | Gift from Dr. A. Nakamura (Institute of Molecular Embryology and Genetics, Kumamoto University) | NA |
| Rabbit α-Bruno 1 | Gift from Dr. M. Lilly (NIH- National Institutes of Child Health and Development) | NA |
| α-Sec23 | ThermoFisher | PA1-069A |
| Mouse α-Tubulin | Deposited to the DSHB by Walsh, C. | AA4.3-s |
| Mouse α-Tubulin | Deposited to the DSHB by Frankel, J./ Nelsen, E.M | 12G10 |
| Rabbit α-GFP | Millipore | AB3080P |
| Rabbit α-Tri-methyl-Histone | Cell Signaling Technology | C42D8 |
| **Oligonucleotides and other sequence-based reagents** | | |
| PCR primers | This study | Table EV1 |
| smFISH probes | This study | Table EV2 |
| **Chemicals, Enzymes and other reagents** | | |
| Phalloidin Alexa Fluor 647 | Life Technologies | A22287 |
| Wheat germ agglutinin | Biotium | CF405S |
| Dylight 488 goat anti-Mouse | ThermoFisher | # 35502 |
| Dylight 550 goat anti-Mouse | ThermoFisher | # 84540 |
| Dylight 650 goat anti-Mouse | ThermoFisher | # 84545 |
| Dylight 650 goat anti-Rabbit | ThermoFisher | # 84546 |
| Schneider's media | ThermoFisher | 21720024 |
| Colcemid | Enzo Life Sciences | 50-200-9161 |
| Halocarbon Oil 700 | Sigma-Aldrich | H8898 |
| O.C.T. Compound | Tissue-Tek | 25608-930 |
| Laemmli Sample Buffer | Bio-Rad | FB2399 |
| TruBlot ULTRA Anti-Mouse Ig HRP | Rockland | 18-8817-31 |
| TruBlot ULTRA Anti-Rabbit Ig HRP | Rockland | 18-8816-31 |
| SuperSignal West Femto Maximum Sensitivity Substrate | ThermoFisher | 34095 |
| TRIzol | ThermoFisher | 15596018 |
| SuperScript IV | Life Technologies | 11766050 |
| PowerTrack SYBR Green Master Mix | Roche Molecular Systems, Inc. | A46110 |
| LysoTracker Deep Red | ThermoFisher | L12492 |

| Reagent/ Resource | Reference or Source | Identifier or Catalog Number |
|---|---|---|
| **Software** | | |
| Imaris image analysis software | Oxford Instruments | NA |
| Fiji | Schindelin et al, 2012 | NA |
| LAS-X | Leica | NA |
| Prism8 | Graphpad | NA |
| BioRender | NA | NA |

## Fly husbandry

*Drosophila melanogaster* stocks were maintained on standard cornmeal agar food at 25 °C. Female flies were put in grape vials and fed yeast paste 2–3 days prior to dissection. Fly stocks obtained from Bloomington *Drosophila* Stock Center: UAS-*mCherry*[RNAi] (BL #35785), *Me31B-GFP* (BL #52530), UAS-*KDEL-RFP* (BL #30910), UAS-*Sec31-RFP* (BL #86532), *tub-a*(V37)-Gal4 (BL #7063), UAS-*bicD*[RNAi] (BL #42929), UAS-*sec16*[RNAi] (BL #53917), UAS-*sec23*[RNAi] (BL #32365), UAS-*sec13*[RNAi] (BL #32468), UAS-*sec31*[RNAi] (BL #32878), UAS-*sar1*[RNAi] (BL #32364), *otu*-Gal4 (BL # 58424). Kyoto *Drosophila* Stock Center: *Cup-YFP* (DGRC 115-161). Vienna *Drosophila* Resource Center: *Rin-GFP* (VDCR #318907). *Oskar-GFP* stock was a generous gift from Dr. G. Gonsalves (Augusta University), and *Tral-RFP*, a generous gift from Dr. St. Johnston (Gurdon Institute at the University of Cambridge). UAS-*mCherry*[RNAi] (BL #35785) was used as a control in all RNAi experiments to account for effects of activated RNAi machinery. All RNAi lines were driven by *tub-a*(V37)-Gal4 (BL #7063) unless otherwise specified. For all experiments, at least 3 biological replicates were used.

## Immunofluorescence

Ovaries were dissected and fixed in 2% PFA in PBS for 10 min. Egg chambers were washed 3× for 10 min in PBST (0.3% Triton X-100), permeabilized and blocked for 2 h in PBS with 1% Triton X-100 and 1% BSA and incubated with primary antibodies overnight at RT with rocking. Followed by 3× 10 min washes in PBST and incubation with fluorescently labeled secondary antibodies (1:1000; DyLight 488, 550, and 650; ThermoFisher). Followed by 3× 10 min washes in PBST and mounted in ProLong Diamond Antifade Mountant (Life Technologies). Antibodies used: Rabbit anti-Sec23 PA1-069A (1:200; ThermoFisher), Rabbit anti-BicD 1B11 deposited to the DSHB by Steward, R. (1:100; Developmental Studies Hybridoma Bank), mouse anti-Cup (1:1000) and mouse anti-Me31B (1:1000) generous gifts from Dr. A. Nakamura (Institute of Molecular Embryology and Genetics, Kumamoto University). Rabbit-anti-Bruno 1 (1:8000) kind gift from Dr. M. Lilly (NIH- National Institutes of Child Health and Development). Actin stain: Phalloidin Alexa Fluor 647 (1:200; Life Technologies). Nuclear membrane stain: wheat germ agglutinin CF405S (1: 200; Biotium).

## Tubulin staining and depolymerization

Ovaries were dissected into BRB80 (0.5M K-PIPES pH 6.8, 2 M MgCl$_2$, and 0.5M K-EGTA) and permeabilized in 1% Triton X-100

in BRB80 without rocking. They were rinsed in BRB80 and fixed with 2% PFA for 10 min. Egg chambers were washed 3× 10 min in PBST followed by 1× 10 min in 2X SSC 10% formamide, and incubated overnight in mouse anti-α-Tubulin AA4.3-s deposited to the DSHB by Walsh, C. and anti-α-Tubulin 12G10 deposited to the DSHB by Frankel, J./Nelsen, E.M (1:100; Developmental Studies Hybridoma Bank). Egg chambers were washed 3× 20 min in PBST and fixed in 4% PFA followed by 3× 10 min washes in PBST preceding a 2 h incubation in a fluorescently labeled secondary (1:1000; Dylight 550 or 650; ThermoFisher). Washed 3× for 10 min in PBST and mounted as previously described. Microtubule depolymerization was achieved by teasing ovaries into Schneider's media (ThermoFisher) supplemented with 10 μM colcemid (Enzo Life Sciences). Ovaries were rocked at RT for 30 min. Control ovaries were dissected into non-supplemented Schneider's media.

## smFISH labeling

Followed method described in (Bayer et al, 2015). In summary, ovaries were dissected and fixed in 4% PFA in PBS for 10 min. Egg chambers were washed 3× 10 min with 2X SSC and pre-hybridized with a 15 min wash in 2X SSC with 10% formamide before being incubated in smFISH probes (1:50) overnight at 37 °C. Egg chambers were washed in pre-warmed 37 °C 2X SSC with 10% formamide 3× 10 min and mounted as previously described. *oskar* 5′ UTR labeled with 46 probes and *oskar* 3′ UTR labeled with 45 probes (Table EV2).

## Microscopy

All imaging was performed on a Leica TCS SP8 Laser Scanning Microscope equipped with a white light laser (470–670 nm), a solid-state laser (405 nm), and a STED 660 nm CW high intensity laser. For all laser scanning confocal imaging, the 63X/1.4 oil objective was used, and optical Z slices were 0.3 μm. For all STED imaging, the 100X/1.4 oil objective was used, optical Z slices were 0.22 μm, and the zoom was 5X. STED samples were prepared from 25 μm ovary sections. Optical sections were acquired using an automated XYZ-piezo stage. All image files were saved as 16-bit data files. Acquisition software: Leica LAS-X.

## Live imaging

Ovaries were dissected into Halocarbon Oil 700 (Sigma-Aldrich) on a #1 coverslip. Egg chambers were adhered by pulling the tissue across the surface as described in (Prasad et al, 2015). FRAP was achieved by bleaching with 100% laser power for 1 min. Recovery was evaluated after 5 min. For all live imaging, the 63X/1.4 oil objective was used. Each movie was acquired at 4X zoom in nurse cells in mid-stage egg chambers. Ten 0.3 μm optical Z sections were acquired every 15 s for 30 min and deconvolved using Leica's Lightning module.

## Sectioning

Ovaries were dissected into 4% PFA in PBS and fixed for 10 min. They were washed 3× 10 min in PBST, 1X for 5 min in 0.1 M glycine pH 3, and stored in 30% sucrose overnight 4 °C. The following day they were flash frozen in O.C.T Compound (Tissue-Tek) before 25 μm sections were prepared on a cryotome.

## Imaging analysis

Identical image acquisition was used for the control and each experimental slide. For all analysis, images were taken from 3 slides each made from a separate fly cross. All images were deconvolved pre-processing using Leica's Lightning module. All images were processed using identical batch parameters and statistics were calculated using Mann–Whitney statistical tests on Prism 8 Software (GraphPad). Figure images were processed using Fiji/ImageJ (NIH) (Schindelin et al, 2012). Quantification was performed using Imaris Microscopy Image Analysis software (Oxford Instruments). Live imaging data was assessed using the Imaris's tracking module. P-bodies and ERES were detected using the Imaris object detection 'Surface' module which allows for object-object statistics. *oskar* mRNA particles were detected with the 'Spot' module as 0.2 μm diameter spots.

Colocalization was determined between 'Spot-Spot', 'Surface-Surface', and 'Spot-Surface' modules. For 'Spot-Spot' less than 0.2 μm distance between two spots was considered colocalization as the software measures from the middle of the spot and each spot had a 0.1 μm radius. For 'Surface-Surface' calculations a shortest distance of less than 0 μm was considered colocalization as this module measures from surface edge to surface edge. For 'Spot-Surface' colocalization, a distance less than 0.1 μm was considered colocalization as the 'Surface' module measures from the edge of a surface to the center of a spot.

## Western blot analysis

Ten ovaries for each genotype were dissected directly into 95 μl of 2X Laemmli Sample Buffer (Bio-Rad) supplemented with 5 μl of BME and immediately mechanically lysed. They were heated at 95 °C for 10 min and centrifuged for 10 min at 10,000 RCF. Lysates were loaded onto a 10% acrylamide gel. Antibodies used: mouse anti-Cup (1:3000), mouse anti-Me31B (1:2000), Tral-RFP detected via Rabbit anti-GFP (1:10,000, Millipore), and Rabbit anti-Tri-methyl-Histone (C42D8) (1:150,000; Cell Signaling Technology). Bands visualized using TrueBlot ULTRA: Anti-Mouse and Anti-Rabbit Ig HRP (1:50,000; Rockland) with SuperSignal West Femto Maximum Sensitivity Substrate (ThermoFisher).

## RNA isolation and RT-qPCR

Whole ovaries were dissected into 4 °C PBS. Ovaries were mechanically lysed in TRIzol (ThermoFisher) to extract total RNA. RNA was washed with ethanol and eluted in RNAse free water. Reverse Transcriptase reactions using 2.5 μg total RNA were performed using Superscript IV kit (Life Technologies). Primers were designed using DRSC FlyPrimerBank and made by Integrated DNA Technologies (Table EV1). RT-qPCR was performed with a Roche Lightcycler 480 (Roche Molecular Systems, Inc.). Each reaction contained 1 μl of cDNA, 4 μl of 10 μM primers, and 5 μl of SYBR Green 1 Master Mix (Roche Diagnostics, Indianapolis, IN). Statistical significance was determined via t-test.

## Stress induction

For cellular stress induction via nutritional deprivation, flies were collected in grape vials and fed yeast paste for 2–3 days. 4 h prior to

dissection, half the flies were moved to new grape vials with no yeast. Ovaries were then fixed and mounted as previously described.

## Autophagy induction and labeling

Flies were collected in grape vials and fed yeast paste for 2 days. To induce autophagy, flies were then starved for 24 h as described in (Barth et al, 2011). Autophagy was labeled by dissecting ovaries directly into Schneider's media. Ovaries were then incubated in 100 μM LysoTracker Deep Red (ThermoFisher) for 2 min to stain acidic organelles. They were then fixed and mounted as previously described.

## Graphics

BioRender was used to prepare Figs. 1A and 6D.

# Data availability

This study includes no data deposited in external repositories.

The source data of this paper are collected in the following database record: biostudies:S-SCDT-10_1038-S44319-024-00344-x.

# Peer review information

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

## Acknowledgements

We thank Dr. A. Spralding (Carnegie Institution for Science), Dr. A. Nakamura, (Riken Center for Developmental Biology), and Dr. M. Lilly (NIH) for the kind gifts of antibodies. We extend thanks to Dr. D. St Johnston (University of Cambridge) and Dr. G. Gonsalves (Augustana University) for the kind gifts of *D. melanogaster* lines. We thank the BDSC Indiana, VDCR Vienna, DGRC Kyoto for providing *D. melanogaster* lines, as well as the TRiP at Harvard Medical School (NIH/NIGMS RO1-GM084947) for the transgenic RNAi stocks. We thank the Bioimaging Facility at Hunter College for access to the Leica TCS SP8 and Imaris -- Image Analysis Software. We thank Dr. P. Feinstein (Hunter College) for allowing us to use the Roche Light-cycler instrument. This work was supported by the National Institute of Health (1SC1GM135132) and the National Science Foundation instrumentation award (1919829) to DPB.

## Author contributions

**Samantha N Milano**: Formal analysis; Investigation; Methodology; Writing—original draft; Writing—review and editing. **Livia V Bayer**: Methodology; Writing—review and editing. **Julie J Ko**: Investigation; Writing—review and editing. **Caroline E Casella**: Investigation; Writing—review and editing. **Diana P Bratu**: Supervision; Funding acquisition; Methodology; Project administration; Writing—review and editing.

Source data underlying figure panels in this paper may have individual authorship assigned. Where available, figure panel/source data authorship is listed in the following database record: biostudies:S-SCDT-10_1038-S44319-024-00344-x.

## Disclosure and competing interests statement

The authors declare no competing interests.

# Expanded View Figures

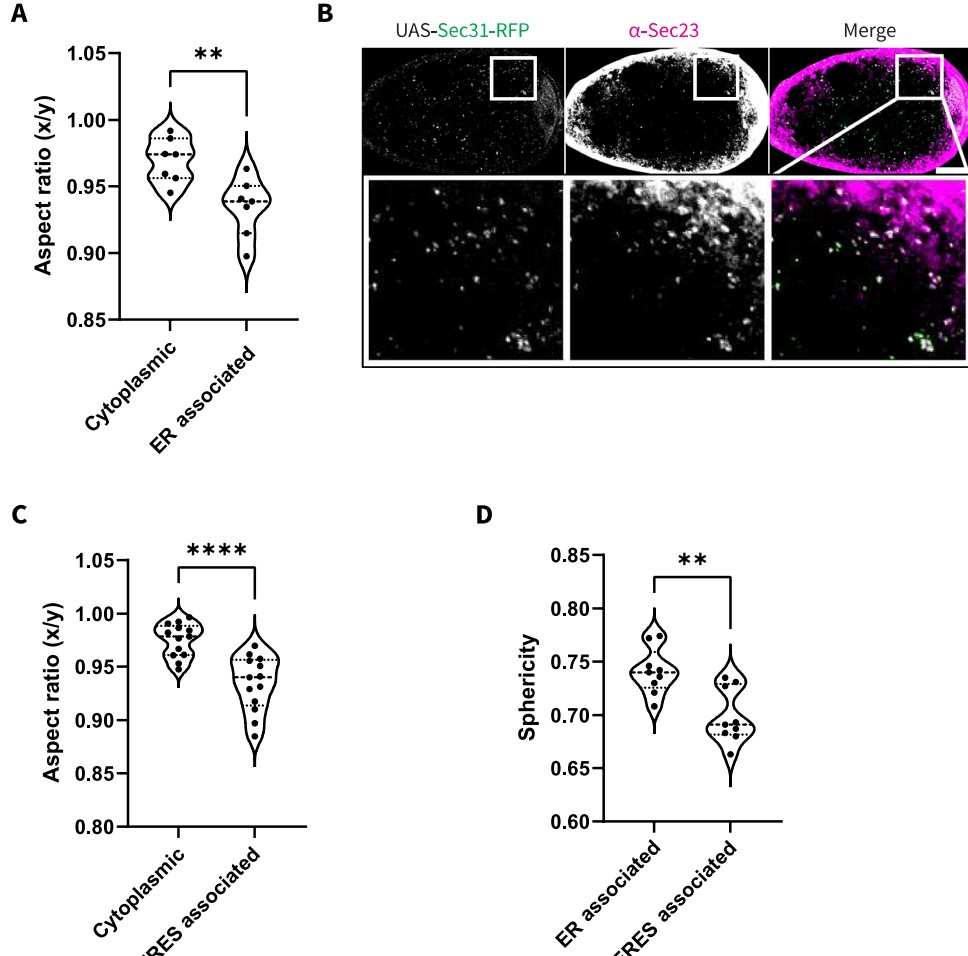

**Figure EV1. P-bodies colocalized with ER exit sites are distinct from cytoplasmic P-bodies.**

(relating to Fig. 1). (**A**) Aspect ratio of cytoplasmic and ER-associated P-bodies (*n* = 7, biological replicates). *P* = 0.0070. (**B**) Co-visualization of Sec31-RFP with Sec23. XY projections of 5 optical Z slices of 0.3 μm. Scale bars are 20 μm and 3.3 μm, respectively, in zoomed inset. (**C**) Aspect ratio of cytoplasmic and ERES-associated P-bodies (*n* = 13, biological replicates). *P* < 0.0001. (**D**) Sphericity measurement comparing ER and ERES-associated P-bodies (*n* = 9, biological replicates). *P* = 0.0028. Significance calculated with Mann–Whitney statistical test. Error bars represent standard deviation. ****\**P* < 0.0001.

Milano et al., 2024

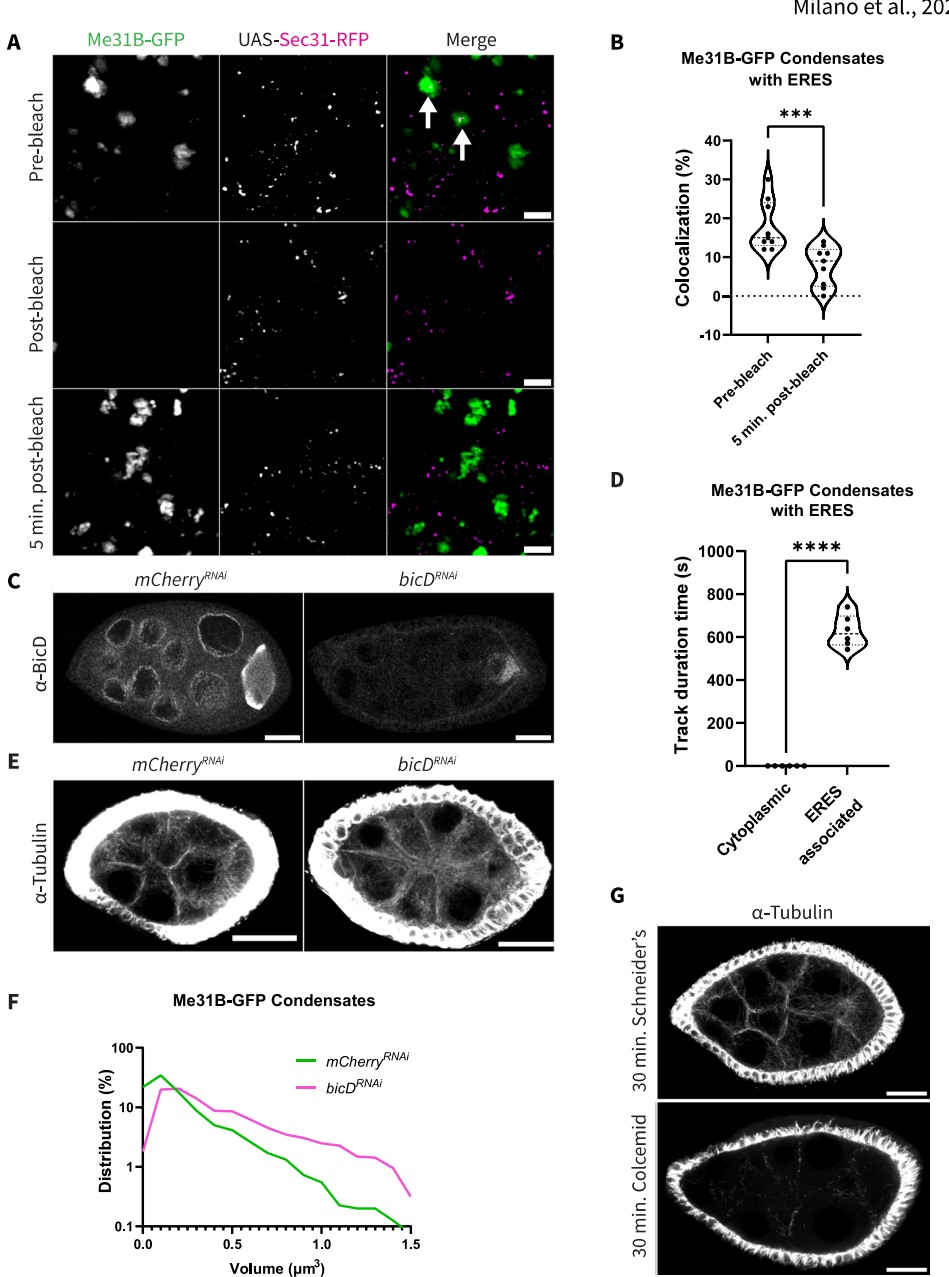

**Figure EV2.  ERES-associated P-bodies are dynamically distinct from cytoplasmic P-bodies and are removed from ERES-associated P-bodies by the microtubule network.**

(relating to Figs. 2 and 3). (A) Co-visualization of Me31B-GFP and Sec31-RFP upon fluorescence recovery after photobleaching. Post-bleach image was acquired immediately after photobleaching. White arrows show ERES-associated P-bodies. Scale bar is 1 µm. (B) Colocalization analysis of Me31B-GFP condensates with Sec31-RFP-labeled ERES (*n* = 9, biological replicates). *P* = 0.0014. Significance calculated with a Mann–Whitney statistical test. (C) BicD expression in *mCherry^RNAi* and *bicD^RNAi* egg chambers. (D) Track duration analysis of Me31B-GFP condensates associated with ERES (*n* = 6, biological replicates). *P* < 0.0001. Significance calculated with a t-test. (E) α-Tubulin labeled microtubules in *mCherry^RNAi* and *bicD^RNAi* egg chambers. (F) Me31B-GFP condensate volume distribution in *mCherry^RNAi* and *bicD^RNAi* backgrounds. X-axis is shown on a log scale. (G) Microtubule integrity in egg chambers after 30 mins incubation in Schneider's media or 10 mM colcemid solution. All images are XY projections of 5 optical Z slices of 0.3 µm. Scale bars are 20 µm. Error bars represent standard deviation. ****P < 0.0001.

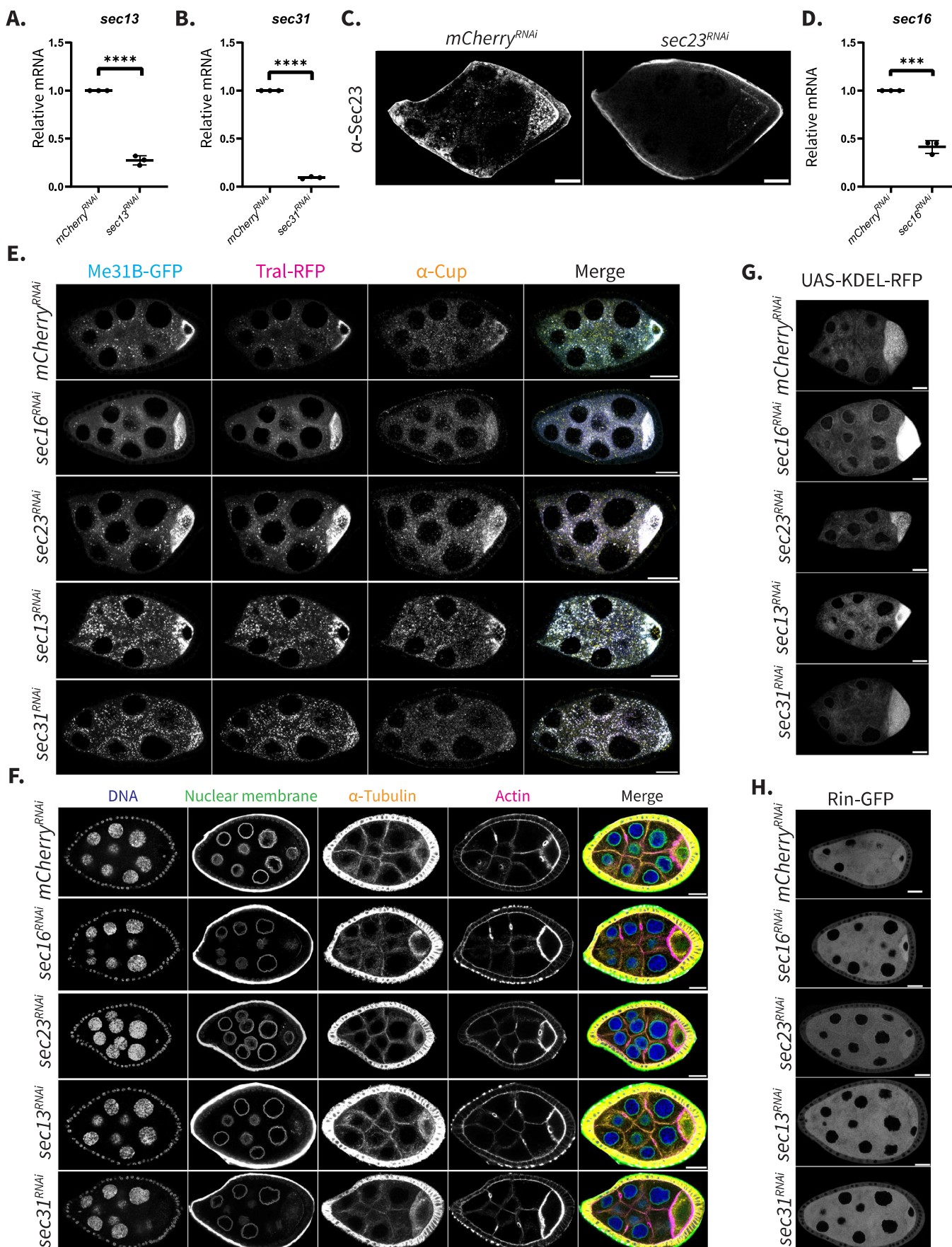

◀  **Figure EV3.  COPII vesicle proteins affect the organization of putative P-body protein condensates.**

(relating to Fig. 4). (**A, B**) Assessing knockdown efficiency of each ERES component via RT-qPCR. ($n = 3$, biological replicates). For $sec13^{RNAi}$ and $sec31^{RNAi}$, $P < 0.0001$. (**C**) Immuno-detection of Sec23 in $mCherry^{RNAi}$ and $sec23^{RNAi}$ egg chambers. (**D**) Assessing knockdown efficiency of $sec16^{RNAi}$ via RT-qPCR, $P = 0.0001$. (**E**) Full egg chamber images of Fig. 3B. (**F**) DNA (DAPI), nuclear membranes (wheat germ agglutinin), microtubules (α-Tubulin), and actin (phalloidin) in COPII component knockdown backgrounds. (**G**) KDEL-RFP visualized in knockdown nurse cells of each ERES component. (**H**) Rin-GFP visualized in each ERES component knockdown background. All images are XY projections of 5 optical Z slices of 0.3 μm. Scale bars are 20 μm. Significance calculated with a t-test. Error bars represent standard deviation. ****$P < 0.0001$.

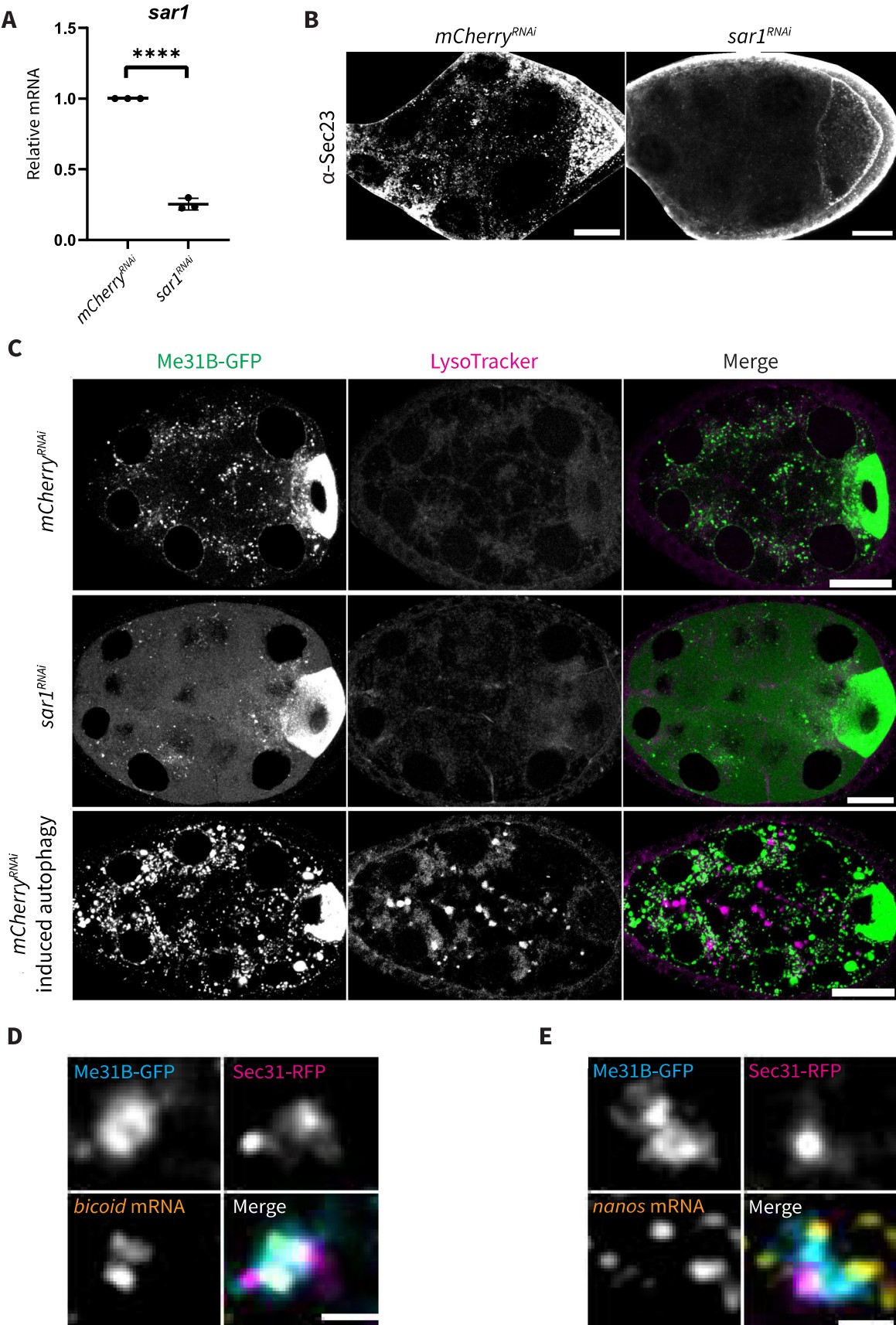

◄ **Figure EV4.  In the absence of ER exit sites, P-body integrity is compromised.**

(relating to Fig. 5). (**A**) Relative levels of *sar1* mRNA in *sar1^RNAi^* egg chambers detected with RT-qPCR. ($n = 3$, biological replicates). $P < 0.0001$. Significance calculated with a t-test. Error bars represent standard deviation from the mean represented by the center bar. ****$P < 0.0001$. (**B**) Sec23 visualized in *mCherry^RNAi^* and *sar1^RNAi^* egg chambers. Scale bars are 20 μm. (**C**) Co-visualization of Me31B-GFP and LysoTracker in *mCherry^RNAi^* and *sar1^RNAi^* egg chambers and in the background of induced autophagy. Scale bars are 20 μm. (**D, E**) Me31B-GFP, Sec31-RFP, and *bicoid* mRNA or *nanos* mRNA visualized with smFISH probes. Scale bars are 1 μm. All images are XY projections of 5 optical Z slices of 0.3 μm.

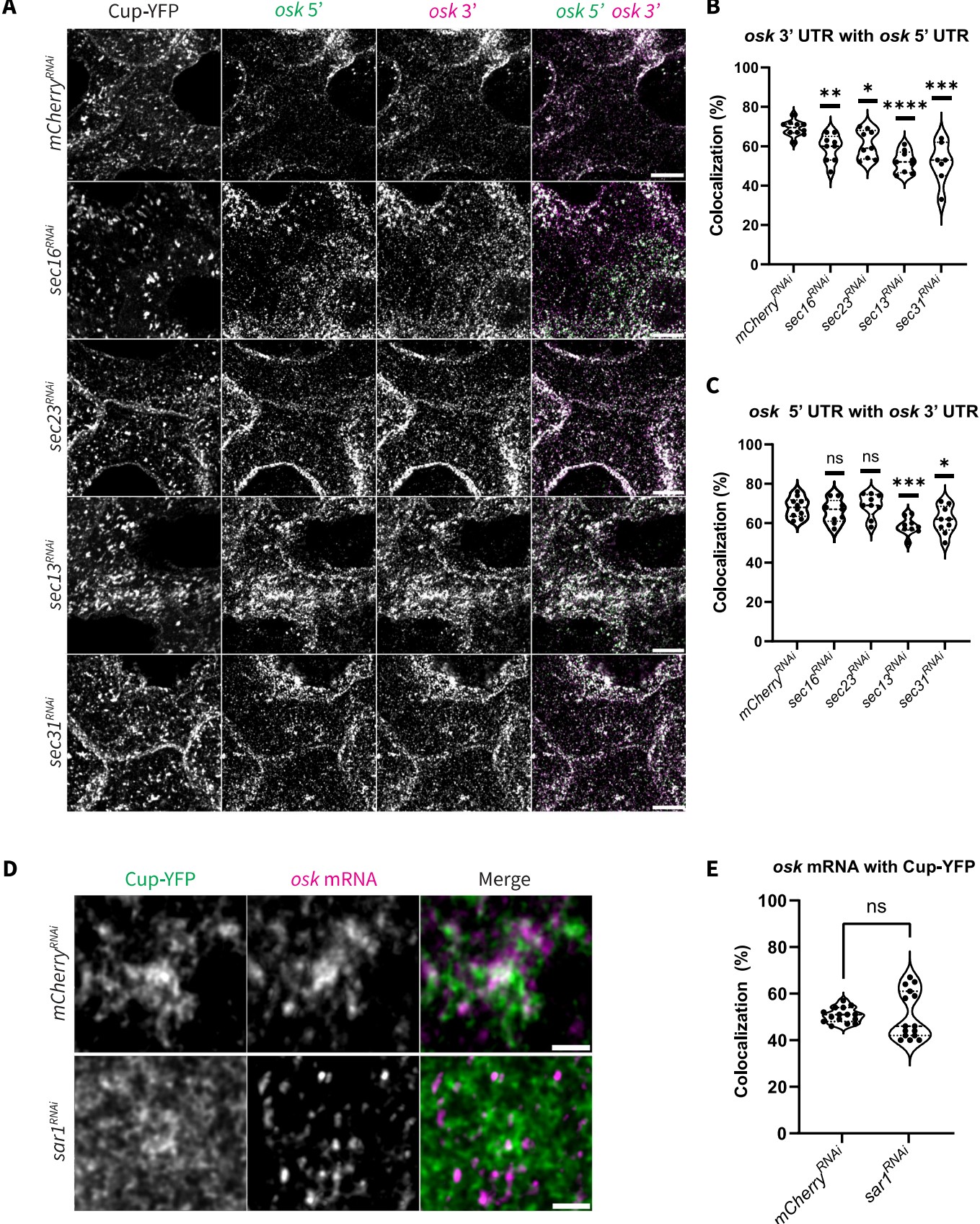

**Figure EV5. Sar1 knockdown leads to attenuation of P-body function.**

(relating to Fig. 6). (**A**) Cup-YFP, *oskar* 5′ UTR, and *oskar* 3′ UTR visualized with respective smFISH probes in each of the COPII component knockdown background. XY projections of 5 optical Z slices of 0.3 μm. Scale bars are 20 μm. (**B**) Colocalization analysis of *oskar* 3′ UTR with *oskar* 5′ UTR ($n = 10$, biological replicates). For *sec16*$^{RNAi}$, $P = 0.0010$. For *sec23*$^{RNAi}$, $P = 0.0104$. For *sec13*$^{RNAi}$, $P < 0.0001$. For *sec31*$^{RNAi}$, $P = 0.0003$. (**C**) Colocalization analysis of *oskar* 5′ UTR with *oskar* 3′ UTR ($n = 10$, biological replicates). For *sec16*$^{RNAi}$ and *sec23*$^{RNAi}$, $P =$ n.s. For *sec13*$^{RNAi}$, $P = 0.0006$. For *sec31*$^{RNAi}$, $P = 0.0451$. (**D**) STED images of a single P-body labeled with Cup-YFP and *oskar* mRNA. XY projections of 3 optical Z slices of 0.22 μm. Scale bars are 2 μm. (**E**) Colocalization analysis of *oskar* mRNA with Cup-YFP ($n = 15$, biological replicates). $P =$ n.s. Significance calculated with a Mann–Whitney statistical test. Error bars represent standard deviation from the mean represented by the center bar. ****$P < 0.0001$.

