## [Peer Review File · EMBO Reports]

The role of ER exit sites in maintaining P-body organization and integrity during *Drosophila melanogaster* oogenesis

Samantha Milano, Livia Bayer, Julie Ko, Caroline Casella, and Diana Bratu

Corresponding author(s): Diana Bratu (bratu@genectr.hunter.cuny.edu)

Review Timeline:

Submission Date:	13th Jul 24
Editorial Decision:	27th Aug 24
Revision Received:	4th Oct 24
Editorial Decision:	8th Nov 24
Revision Received:	8th Nov 24
Accepted:	15th Nov 24

Transaction Report:

Dear Prof. Bratu

Thank you for the submission of your research manuscript to our journal. We have now received the full set of referee reports that is copied below.

As you will see, the referees acknowledge that the findings are potentially interesting, but they also raise a number of overlapping concerns. Clearly, the analysis of P-body physical shape and the conclusions derived from this need to be substantiated and the analysis extended beyond indicators such as 'surface texture'. Referee 3 suggests an interesting alternative hypothesis, i.e., that Sec16 and Sar1 per se and not the ERES affect P-body attachment. That said, I agree with an important concern from referee 1: depletion of ERES proteins will have a rather pleiotropic effect on the cells, questioning the proposed direct link to P-body state. A functional separation of a general effect of ERES disruption on the cell and a specific effect on P-bodies might be difficult but effects on protein transport and other ERES-mediated functions such as autophagy need to be analysed, documented and discussed.

From the referee comments, it is clear, that a significant revision will be required to substantiate your findings, but I would be willing to give you the chance to revise your manuscript with the understanding that the referee concerns (as detailed above and in their reports) must be fully addressed and their suggestions taken on board. Please address all referee concerns in a complete point-by-point response. Acceptance of the manuscript will depend on a positive outcome of a second round of review. It is EMBO Reports policy to allow a single round of revision only and acceptance or rejection of the manuscript will therefore depend on the completeness of your responses included in the next, final version of the manuscript.

We realize that it is difficult to revise to a specific deadline. In the interest of protecting the conceptual advance provided by the work, we recommend a revision within 3 months (November 27). Please discuss the revision progress ahead of this time with the editor if you require more time to complete the revisions.

I am also happy to discuss the revision further via e-mail or a video call, if you wish.

*******IMPORTANT NOTE:**

We perform an initial quality control of all revised manuscripts before re-review. Your manuscript will FAIL this control and the handling will be delayed IN CASE the following APPLIES:

- 1) A data availability section providing access to data deposited in public databases is missing. If you have not deposited any data, please add a sentence to the data availability section that explains that.
- 2) Your manuscript contains statistics and error bars based on $n=2$. Please use scatter blots in these cases. No statistics should be calculated if $n=2$.

When submitting your revised manuscript, please carefully review the instructions that follow below. Failure to include requested items will delay the evaluation of your revision. *****

- 1) a .docx formatted version of the manuscript text (including legends for main figures, EV figures and tables). Please make sure that the changes are highlighted to be clearly visible.
- 2) individual production quality figure files as .eps, .tif, .jpg (one file per figure). Please download our Figure Preparation Guidelines (figure preparation pdf) from our Author Guidelines pages <https://www.embopress.org/page/journal/14693178/authorguide> for more info on how to prepare your figures.
- 3) a .docx formatted letter INCLUDING the reviewers' reports and your detailed point-by-point responses to their comments. As part of the EMBO Press transparent editorial process, the point-by-point response is part of the Review Process File (RPF), which will be published alongside your paper.
- 4) a complete author checklist, which you can download from our author guidelines (). Please insert information in the checklist

that is also reflected in the manuscript. The completed author checklist will also be part of the RPF.

5) Please note that all corresponding authors are required to supply an ORCID ID for their name upon submission of a revised manuscript (). Please find instructions on how to link your ORCID ID to your account in our manuscript tracking system in our Author guidelines

()

6) We replaced Supplementary Information with Expanded View (EV) Figures and Tables that are collapsible/expandable online. A maximum of 5 EV Figures can be typeset. EV Figures should be cited as 'Figure EV1, Figure EV2' etc... in the text and their respective legends should be included in the main text after the legends of regular figures.

7) Please include a dedicated "Data Availability" section at the end of the Methods (suggested wording: "The [structural coordinates | microarray | mass spectrometry] data from this publication have been deposited to the [name of the database] database [URL] and assigned the identifier [accession | permalink | hashtag]."). Should this not apply, this should still be stated as "This study includes no data deposited in external repositories."

Additional information on source data and instruction on how to label the files are available .

10) Figure legends and data quantification:

- the name of the statistical test used to generate error bars and P values,
 - the number (n) of independent experiments (please specify technical or biological replicates) underlying each data point,
 - the nature of the bars and error bars (s.d., s.e.m.)
- If the data are obtained from n {less than or equal to} 5, show the individual data points in addition to the SD or SEM.
- If the data are obtained from n {less than or equal to} 2, use scatter blots showing the individual data points.

11) Our journal encourages inclusion of *data citations in the reference list* to directly cite datasets that were re-used and obtained from public databases. Data citations in the article text are distinct from normal bibliographical citations and should directly link to the database records from which the data can be accessed. In the main text, data citations are formatted as follows: "Data ref: Smith et al, 2001" or "Data ref: NCBI Sequence Read Archive PRJNA342805, 2017". In the Reference list, data citations must be labeled with "[DATASET]". A data reference must provide the database name, accession number/identifiers and a resolvable link to the landing page from which the data can be accessed at the end of the reference. Further instructions are available at .

12) All Materials and Methods need to be described in the main text using our 'Structured Methods' format, which is required for all research articles. According to this format, the Methods section includes a Reagents and Tools Table (listing key reagents, experimental models, software and relevant equipment and including their sources and relevant identifiers) followed by a

Methods and Protocols section describing the methods using a step-by-step protocol format. The aim is to facilitate adoption of the methodologies across labs. More information on how to adhere to this format as well as a downloadable template (.docx) for the Reagents and Tools Table can be found in our author guidelines:
<https://www.embopress.org/page/journal/14693178/authorguide#structuredmethods>.

An example of a Method paper with Structured Methods can be found here:
<https://www.embopress.org/doi/10.15252/msb.20178071>.

13) As part of the EMBO publication's Transparent Editorial Process, EMBO Reports publishes online a Review Process File to accompany accepted manuscripts. This File will be published in conjunction with your paper and will include the referee reports, your point-by-point response and all pertinent correspondence relating to the manuscript.

Yours sincerely,

Referee #1:

The idea that ERES control p-body biology is interesting and an exciting avenue. As the authors point out, previous reports have shown an interaction between COPII proteins and p-body components, a subsequent step would be to add mechanistic insight into this interaction.

The authors deplete COPII / ERES proteins and show an effect on p-body organization. There is no clear evidence that this effect is direct or because of any one of the dramatic indirect changes one would expect with ERES perturbation, including altered autophagy - a process thought to affect p-body biology.

The authors use 'surface texture' as an indicator of condensate physical state. The ERES is crowded with tubulovesicular membranes and one would reasonably expect any condensate in the area to appear less spherical merely because it is filling the spaces between these membranes. Better evidence is required to suggest that the condensate has a different physical state. Further, at these size scales and resolutions (even with the STED images), it is impossible to distinguish between a cluster of p-bodies forming or a single large p-body. Distinguishing between these two is very important for the interpretation of shape and organization of p-bodies.

The model presented is confusing. The authors simultaneously suggest that ERES P-bodies are 'more mature than their cytoplasmic counterparts' (as stated in the abstract), while also suggesting that p-bodies are nucleated and form at the ERES (as in the graphical abstract).

Is there any difference in composition between the ERES-associated p-bodies and cytoplasmic p-bodies? If not, then the simplest model is that all p-bodies are the same, but if they encounter ERES, they are temporarily held there.

Is there any effect on ERES morphology and function in secretion if p-bodies are eliminated, or prevented from being recruited to ERES?

Referee #2:

This work characterizes two subpopulations of Processing bodies (P-bodies) in *Drosophila* egg chambers based on their physical properties. The work proposes that ERES associated P-bodies have a functional role in the formation of cytoplasmic P-bodies and that ERES components are important for the physical state of ERES P-bodies. They also present data that ERES P-bodies have a biological function through the regulation of axis-determining mRNAs. This data is not presented as central to the manuscript, but if proven, is likely the most impactful aspect of the research. For the majority of the work, the experiments are rigorous in design and analysis with the data being presented clearly and at the highest level. The writing is informative and motivated the reader, both expert and general. The language is concise and logical. Conclusions are supported by the data in most all cases with some additional clarity necessary in a few specific places.

MAJOR COMMENTS

- 1) It is unclear what stage and where in the egg chamber images are from (Figure 1-3). This information should be included in the text or images. A schematic would be helpful to inform the reader as to when and where the presented data is from.
- 2) Figure 4 (and Figure EV4) is not of the same high-quality as the rest of the figures. Quantification is necessary for this work to reach the same level as the rest of the data. Image quality is noticeably different than other figures. The presentation is not consistent (whole egg chambers being shown with small insets) with earlier figures. One of the most exciting and important findings is currently in the supplemental data and should be moved to the main manuscript (Fig. EV4E, G, H, F showing oskar (osk) mRNA localization and translation).

MINOR COMMENTS

Abstract/Introduction:

- 1) "More mature" is not necessarily shown. In order to establish this more detailed analysis of the condensates is required. Stating that the P-bodies are different rather than "mature" would be more accurate.
- 2) "One method by which this silenced transport is achieved is via processing bodies (P-bodies)" is in need of a reference.

Figure 1

- 3) It would be helpful to add arrows highlighting examples of associated and not associated P-bodies in the 8X zoom images.
- 4) Since colocalization is important through the work, it would be important to briefly explain how colocalization analysis was completed in the the main text (in addition to the very clear explanation in the methods).
- 5) Is sphericity impacted by size? Perhaps aspect ratio, which is not impacted by size, could be shown in a few cases for comparison.
- 6) It would be helpful to have a longer explanation for panel and reference(s) for the rough vs smooth discussion in panel G and the pixel intensity SD analysis in panel H.
- 7) When comparing Figure 1A to 1D and Figure 1C to 1F, the volumes are an order of magnitude different. It would be helpful to comment on this.
- 8) It would be good to add a more zoomed out image to G, consistent with the top panels of A and D.
- 9) The statement, "...'surface texture' as an indicator of condensate physical state (Fig. 1G)" would benefit from a reference and/or further expatiation.
- 10) The statement, "As condensates can mature and 'harden' over time, this may indicate that these P-bodies are older and forming gel-like condensate states, perhaps suggesting that these condensates are older than cytoplasmic P-bodies (10,11)." Should be rephrased for clarity. Is it known that there is causation between being "hard" and older in this in vivo setting?
- 11) In order to test the physical state of the P-bodies in panel G, a FRAP would be a nice addition (although not required), especially since Figure 2B shows they are not very dynamic.

Figure 2

- 12) In Figure 2A, two panels show 90 seconds.
- 13) Figure 2C appears to be reversed with cytoplasmic less linear than ERES associated.
- 14) The statement, "...suggesting that P-bodies move along microtubules via the Egalitarian/BicD/Dynein complex (38)." - is this referring to in the nurse cells or the oocyte? It has also been previously shown.
- 15) Could cytoplasmic P-bodies be "building" the ERES P-bodies? Is it possible to shown that MTs are pulling out cytoplasmic P-bodies versus MTs bringing in cytoplasmic P-bodies. Could this be discussed? (Live imaging showing budding or fusion would help to clarify this, but is not required.)
- 16) Is Figure 2H necessary since Figure 2J effectively shows the same data?

Figure 3

- 17) Could blocking with RNAi lines cause proteins to accumulate in the ER and lead to increased P-body size? If this is possible perhaps it could be discussed.
 - 18) Why was anti-CUP used for image and Cup-YFP used for analysis (Figure 3B compared to 3H and 3I).
 - 19) In Figure 3B, anti-Cup appears in the nucleus and there is significant amount of Cup detected without Me31B and 20) Tral. Does this suggest that Cup might not be in these P-bodies?
- Future 1 and 2 shows nicely that there are two populations of P-bodies. In Figure 3B - 3I, the images and analysis appears to be on all P-bodies in these cells. Could this impact the analysis and conclusions? If possible, please comment on this.

Figure 4

- 21) The statement, "As many maternal mRNA's are recruited into P-bodies...", needs a reference.
- 22) In Figure 4E, the egg chambers appear to be different stages based on the nucleus position in the oocyte.
- 23) Figure 4F does not seem central to the story and could be moved to EV4. It would also be helpful to show a stressed, mcherryRNAi control (or any non-sar1RNAi control).
- 24) In Figure EV4E and F, P-bodies (marked with Me31B and Cup) look very different. It would be helpful to comment on this.
- 25) In Figure EV4E and F, the osk mRNA distribution appears different. (E looks good, F needs to be consistent).
- 26) In Figure EV4I, it is surprising that Osk protein forms large and distinct particles in the oocyte and is not translated in the nurse cells. Could this be discussed?

Referee #3:

The manuscript by Milano et al describes a role for ERES proteins in the localization, biogenesis and/or maintenance of P-bodies. The work is in general well written and the experiments are conclusive. I think the topic is timely, given that the notion that LLPS regulates the formation of ERES and other organelles of the secretory pathways. Therefore, I am generally in favor of publication of this work, but I have some questions and comments that should be addressed before I can fully recommend this paper for publication.

1- My main concern is with the experiments in Figure 3. Firstly, the authors use Sar1 knockdown to disrupt ERES. This is correct, but a knockdown of Sec16 should also disrupt ERES formation. Sec16 is the main organizer of ERES biogenesis. In *Drosophila*, ERES form on cup-shaped structures and Sec16 is an organizer of these structures. What is the reason for the difference between Sar1 and Sec16? Is it that Sar1 plays a much more fundamental role in P-body formation than Sec16 and other ERES components? I think the authors should either address this experimentally, or offer some balanced discussion on this matter. In any case, they should avoid making this distinction between Sar1 and Sec16. The distinction is correct for the more downstream elements such as Sec23, Sec13 and Sec31. These do not affect the formation of ERES, which explains their milder effects.

An alternative interpretation of these results might be that Sec16 per se (and not the ERES) might play an important role in attaching P-bodies to ERES and that Sar1 plays an even more fundamental role. This more fundamental role of Sar1 might also explain why its knockdown affects even the cytoplasmic P-bodies (those not associated with the ERES). My view might well be wrong, but then the authors should explain why. Without such explanation, I am (as a reader) left with too much uncertainty.

2- Is there a reason why the authors decided not to use a FRAP assay to assess the liquidity of the P-bodies? Unless I am missing something, I think that a FRAP assay will inform more accurately about the liquid state of the P-body than the roughness of its surface. I also think that this is a simple experiment that could be easily done in the course of a revision.

3- Could the authors elaborate a little bit on how they think that microtubules would "pull" on a P-body. An effect of microtubules on motility of P-bodies might be the consequence of a change in cytoplasmic viscosity. This is not a request for an experiment, but I think it would be nice for the reader to read some discussion on this matter.

Minor comments:

- It would help the general reader if the authors could point out examples for ER-associated and cytoplasmic P-bodies.
- It is also not very clear how the authors determined whether a given P-body is ER-associated.
- It would be interesting to report on the instantaneous speed of P-bodies in addition to the MSD and the linearity of the track. I would imagine that cytoplasmic P-bodies are faster because they are less obstructed? The speed cannot be calculated by dividing the distance covered by time. It should be the instantaneous speed that is measured.

Dear Editor and Referees,

Thank you for the opportunity to respond to your comments and implement your recommendations. We believe our revised manuscript has gained a lot of clarity, impact, and depth due to your thorough reviews. Please find below our individual responses to all the points raised by each of you. We hope you find our responses suitable to alleviate the concerns raised, and that we satisfactorily addressed all the suggestions provided.

Referee #1:

The idea that ERES control p-body biology is interesting and an exciting avenue. As the authors point out, previous reports have shown an interaction between COPII proteins and p-body components, a subsequent step would be to add mechanistic insight into this interaction. The authors deplete COPII / ERES proteins and show an effect on p-body organization. There is no clear evidence that this effect is direct or because of any one of the dramatic indirect changes one would expect with ERES perturbation, including altered autophagy - a process thought to affect p-body biology.

We acknowledge that ERES depletion through RNAi knockdown may lead to various pleiotropic effects. However, our analysis revealed no significant changes in microtubule organization, stress granule formation, nor detected any gross alterations to the ER membrane - the dramatic changes expected when disruptions of the secretion functions of the ER occur (Figure EV3F-H). While autophagy could possibly mask the interpretation of our results, all ERES component knockdown egg chambers that we analyzed, except Sar1, progressed through oogenesis. Therefore, we do not believe these egg chambers were affected by autophagy. For sar1^{RNAi} egg chambers, we tested whether they underwent autophagy at the developmental stages at which we did our analysis, via visualizing accumulation of acidic organelles using LysoTracker. We added the following text into the manuscript to describe this experiment (page #16):

“As the Sar1 knockdown induced cell death at stage 8, we were concerned that the altered P-body distribution might be a consequence of the early-stages of apoptosis. To investigate this possibility, we employed LysoTracker to monitor autophagy but observed no signs of autophagic activity during the stages analyzed. To further rule out the role of autophagy in the diffuse P-body phenotype, we examined P-bodies in egg chambers from flies subjected to starvation to induce autophagy, as described by Barth et al. (2011). Notably, in this context, P-bodies appeared larger and more punctate, in stark contrast to the diffuse morphology observed in the Sar1 knockdown (Fig. EV4C). Collectively, these results indicate that autophagy is not responsible for the diffusion of P-body proteins.”

The authors use 'surface texture' as an indicator of condensate physical state. The ERES is crowded with tubulovesicular membranes, and one would reasonably expect any condensate in the area to appear less spherical merely because it is filling the spaces between these membranes. Better evidence is required to suggest that the condensate has a different physical state.

While we too believe that it is possible for sphericity to be influenced by sheer force of ER membranes, we utilized the parameter of 'surface roughness' to circumvent this possibility. Surface roughness detects the distribution of fluorophores within a condensate. Even if a liquid-like condensate was against a solid surface and deformed to become less spherical, the fluorescent markers would still flow within the volume and the surface roughness would still be

smoother than a solid-like P-body. We have reworded the surface roughness section to reflect this explanation more clearly (page #7):

“Texture assesses the distribution of fluorophores within a cross section of a condensate, thus providing a read out of the internal structure of a P-body. ‘Rougher’ P-bodies indicate more solid/gel-like condensates as fluorescence signal is distributed in clumped structures, while ‘smoother’ P-bodies indicate more liquid-like condensates, with more evenly dispersed fluorescence signal (Shiina et al; 2019).”

Similarly, to ensure that the tubulovesicular structures of the ER are not affecting the sphericity measurements, we add (page #7):

“Given the possibility that the sphericity of condensates might be affected by the crowded, tubulovesicular nature of the ER, we compared the sphericity of ER-associated P-bodies and ERES-associated P-bodies. As all P-bodies in both datasets were colocalized with the ER membrane, we believe that any effects of crowding are eliminated and ERES-associated P-bodies are less spherical than ER-associated P-bodies, suggesting that they may be more gel-like (Fig. EV1D).”

Further, at these size scales and resolutions (even with the STED images), it is impossible to distinguish between a cluster of p-bodies forming or a single large p-body. Distinguishing between these two is very important for the interpretation of shape and organization of p-bodies.

Detecting individual P-bodies is crucial yet defining a “single” P-body structure remains a challenge. The delineation of a P-body's boundary is influenced by the marker employed, particularly given the absence of a membrane enclosure. Our Me31B-GFP imaging data, obtained from both live and fixed egg chambers using confocal and super-resolution techniques, reveal a diverse range of sizes that encompass both individual and clustered P-bodies. For our analysis, we include fluorescent “objects” ranging from 50 nm to 2 microns in size.

With super-resolution imaging we can resolve the distance between two P-bodies as short as 8 nm. When two P-bodies are below this resolution limit and touching, we consider them to become a larger, single P-body by virtue of their liquid-like characteristics.

We propose that characteristics such as “shape” (spherical- indicating liquid versus reticulated- indicating non-liquid) are independent of P-body size, whereas their organization (clustered versus dispersed) is determined by the cellular context in which these P-bodies are found.

The model presented is confusing. The authors simultaneously suggest that ERES P-bodies are 'more mature than their cytoplasmic counterparts' (as stated in the abstract), while also suggesting that p-bodies are nucleated and form at the ERES (as in the graphical abstract).

We agree that there is this discrepancy in our explanation. To clarify, we removed the word ‘mature’, which in this context was meant to indicate that they are ‘older’/‘forming first’, and reformulated the explanation of our proposed mechanism (page #19):

“We propose a model where ERES formation contributes to the organization of larger, gel-like P-bodies, which serve as nucleating hubs from which smaller, liquid-like, P-bodies can bud off via the cytoskeletal network and become more mobile.”

We separated the data in Figure 2 into two different figures (Figure 2 and 3-New) to allow for a more elaborate presentation of the data and justification of our proposed mechanism.

Is there any difference in composition between the ERES-associated p-bodies and cytoplasmic p-bodies? If not, then the simplest model is that all p-bodies are the same, but if they encounter ERES, they are temporarily held there.

Assessing the differential composition of P-bodies within an egg chamber has become a significant focus in the field. However, biochemical approaches have not yet succeeded in elucidating any distinct compositions. To address this gap, we plan to undertake a large-scale multiplex imaging screen using endogenously labeled P-body factors in conjunction with multiple maternal mRNAs, which will provide us with a comprehensive understanding of the P-body population. This is however outside the scope of this paper.

While we believe that ERES-P-bodies have unique compositions that are yet to be delineated, we argue that they can also be differentiated based on other condensate features, such as structure, dynamics, and function. We have reworked the relevant section for clarity (page #11) and included a figure illustrating the extended duration of interactions between ERES P-bodies and ERES (Figure EV2D).

Is there any effect on ERES morphology and function in secretion if p-bodies are eliminated, or prevented from being recruited to ERES?

*Previous research has demonstrated that in *Tral* mutants—an important P-body protein—*Gurken* protein secretion is disrupted and *Sar1* exhibits abnormal localization. However, there is a notable gap in understanding how P-bodies regulate ERES function, warranting a more in-depth investigation into this relationship. We have added the following statement (page# 4):*
*“These studies further revealed that *Tral* directly binds the mRNAs of two COPII proteins: *Sar1* and *Sec13*, and in the absence of *Tral* protein, ERES are aberrantly localized, and secretion is disrupted indicating that P-bodies play a role in ERES regulation (Wilhelm et al, 2005).*

Referee #2:

Thank you for the organization of your comments.

This work characterizes two subpopulations of Processing bodies (P-bodies) in *Drosophila* egg chambers based on their physical properties. The work proposes that ERES associated P-bodies have a functional role in the formation of cytoplasmic P-bodies and that ERES components are important for the physical state of ERES P-bodies. They also present data that ERES P-bodies have a biological function through the regulation of axis-determining mRNAs. This data is not presented as central to the manuscript, but if proven, is likely the most impactful aspect of the research.

For the majority of the work, the experiments are rigorous in design and analysis with the data being presented clearly and at the highest level. The writing is informative and motivated the reader, both expert and general. The language is concise and logical. Conclusions are supported by the data in most all cases with some additional clarity necessary in a few specific places.

MAJOR COMMENTS

1) It is unclear what stage and where in the egg chamber images are from (Figure 1-3). This information should be included in the text or images. A schematic would be helpful to inform the reader as to when and where the presented data is from.

To help guide the non-fly readers, we included a schematic of the developmental stages of an ovariole in Figure 1A that highlights the nurse cell region where imaging was conducted. We have noted the developmental stages across all figure legends.

2) Figure 4 (and Figure EV4) is not of the same high-quality as the rest of the figures. Quantification is necessary for this work to reach the same level as the rest of the data.

Figure 5 (previous Figure 4) is now represented with better quality image panels.

We added histograms to the first two image panels (new Figure 5B-C and E-F) to highlight the alteration in cytoplasmic distribution. We also added quantification of the oskar mRNA particles in the sar^{RNAi} background (Figure 5J).

Image quality is noticeably different than other figures. The presentation is not consistent (whole egg chambers being shown with small insets) with earlier figures.

We have re-processed some of the image data sets and, in some cases, manually removed the follicle cells where the fluorescence signal overwhelmed the information in the nurse cells; we believe the germline data is thus presented more clearly. We chose to illustrate the expression patterns in whole egg chambers in the sar^{RNAi} background as we wish to highlight the difference in protein and mRNA localization in the oocyte as well as the distribution in the nurse cells. We included zoomed images of these egg chambers in Figure 6H.

One of the most exciting and important findings is currently in the supplemental data and should be moved to the main manuscript (Fig. EV4E, G, H, F showing oskar (osk) mRNA localization and translation).

Yes, we agree and have moved the data into Figure 6.

MINOR COMMENTS

Abstract/Introduction:

1) "More mature" is not necessarily shown. In order to establish this more detailed analysis of the condensates is required. Stating that the P-bodies are different rather than "mature" would be more accurate.

*We agree and thus have reworked the wording to better articulate our observations (page#2):
"Employing a combination of live and super-resolution imaging, we found that P-bodies associated with ER exit sites are larger and less mobile than cytoplasmic P-bodies, indicating that they constitute a distinct class of P-bodies."*

2) "One method by which this silenced transport is achieved is via processing bodies (P-bodies)" is in need of a reference.

References have been added (page #3):

"One method by which this silenced transport may be achieved is via processing bodies (P-bodies) (Weil et al., 2012, Bayer et al., 2024)."

Weil, T.T., Parton, R.M., Herpers, B., Soetaert, J., Veenendaal, T., Xanthakis, D., Dobbie, I.M., Halstead, J.M., Hayashi, R., Rabouille, C., Davis, I., 2012. Drosophila patterning is established by differential association of mRNAs with P bodies. Nat Cell Biol 14, 1305–1313.

Bayer, L.V., Milano, S.N., Bratu, D.P., 2024. The mRNA dynamics underpinning translational control mechanisms of Drosophila melanogaster oogenesis. Biochemical Society Transactions BST20231293.

Figure 1

3) It would be helpful to add arrows highlighting examples of associated and not associated P-bodies in the 8X zoom images.

Both arrows and arrowheads have been added to distinctly indicate each population within the Merge panels in Figure 1B, E and Figure 3B, E.

4) Since colocalization is important through the work, it would be important to briefly explain how colocalization analysis was completed in the main text (in addition to the very clear explanation in the methods).

We have added a more nuanced explanation in the text (page#6):

“To differentiate between cytoplasmic P-bodies and ER-associated P-bodies we used Imaris image analysis software to detect P-body and ER surfaces. We binned all detected P-bodies that have shortest distances to the ER $\leq 0 \mu\text{m}$ (touch or overlap) as ER-associated P-bodies and any P-bodies with a shortest distance to the ER $> 0 \mu\text{m}$ as a cytoplasmic P-body.”

5) Is sphericity impacted by size? Perhaps aspect ratio, which is not impacted by size, could be shown in a few cases for comparison.

We believe sphericity is a more accurate representation of our data as it calculates deviation from a perfect sphere and thus provides a better assessment of subtle changes in morphology, however we have also included an assessment of aspect ratio in Figure EV1A, C.

6) It would be helpful to have a longer explanation for panel and reference(s) for the rough vs smooth discussion in panel G and the pixel intensity SD analysis in panel H.

We expanded on our explanation and provided additional references (page #7-8):

“To address this question, we employed STED super-resolution microscopy to visualize condensate ‘texture’ to provide insight into a condensate’s physical state (Fig. 1G). Texture assesses the distribution of fluorophores within a cross section of a condensate, thus providing a readout of the internal structure of a P-body. ‘Rougher’ P-bodies indicate more solid/gel-like condensates as fluorescence signal is distributed in clumped structures, while ‘smoother’ P-bodies indicate more liquid-like condensates, with more evenly dispersed fluorescence signal (Shiina et al; 2019). To quantify fluorescence distribution within a P-body, we utilized standard deviation (SD) of pixel intensity (Irgen-Gioro et al., 2022). This technique requires assessing the intensity of each pixel within the boundary of a P-body and obtaining a SD which indicates the range of pixel intensities within the condensate.”

Shiina, N., 2019. Liquid- and solid-like RNA granules form through specific scaffold proteins and combine into biphasic granules. Journal of Biological Chemistry 294, 3532–3548.

Irgen-Gioro, S., Yoshida, S., Walling, V., Chong, S., 2022. Fixation can change the appearance of phase separation in living cells. eLife 11, e79903.

7) When comparing Figure 1A to 1D and Figure 1C to 1F, the volumes are an order of magnitude different. It would be helpful to comment on this.

We added the following text (page # 7):

“Interestingly, the average volume of ER-associated P-bodies is significantly larger than those associated with just ERES (0.973 μm^3 compared to 0.324 μm^3). This may be due to the differential localization of ERES and the whole ER membrane. ERES are centrally localized in the nurse cell cytoplasm while the ER membrane in general extends to regions of the nurse cell cytoplasm where P-bodies congregate and coalesce. For example, as the (-) ends of microtubules are oriented towards the periphery of the nurse cells, many P-bodies accumulate there (Bayer et al, 2024; Theurkauf and Hazelrigg, 1998). Similarly, we found the ring canal areas which allow for the cytoplasm to communicate between nurse cells, have an abundance of P-bodies while they attempt to move through a small volume, thus becoming bottlenecked and clustered into larger entities.”

8) It would be good to add a more zoomed out image to G, consistent with the top panels of A and D.

Although it would be visually ideal to have a consistent image for this panel, our STED imaging required a higher zoom level to meet Nyquist sampling criteria. This means the pixel size must be at least 2.3 times smaller than the object being resolved.

9) The statement, "...'surface texture' as an indicator of condensate physical state (Fig. 1G)" would benefit from a reference and/or further expatiation.

See the response to point #6. We included a reference as well as provided further explanation.

10) The statement, "As condensates can mature and 'harden' over time, this may indicate that these P-bodies are older and forming gel-like condensate states, perhaps suggesting that these condensates are older than cytoplasmic P-bodies (10,11)." Should be rephrased for clarity. Is it known that there is causation between being "hard" and older in this in vivo setting?

We do not know the direct cause of the 'hardening' of these cytoplasmic bodies. We speculate that overtime intra-condensate interactions can lead to the formation of fibers thus hardening the condensates. This phenomenon has been shown in vitro but it is difficult to ascertain direct causation within an in vivo system. We reworded this statement for better clarity (page #8):

“This may indicate that these P-bodies are taking on gel-like condensate states, and as condensates were shown to ‘harden’ over time in vitro, it may also suggest that these condensates are more mature (Lin et al, 2015; Garaizar et al, 2022).”

11) In order to test the physical state of the P-bodies in panel G, a FRAP would be a nice addition (although not required), especially since Figure 2B shows they are not very dynamic.

Since we do not have access to a FRAP system, we relied on our super resolution imaging to perform our analysis. While we decided to employ a modified version of FRAP to capture overall condensate dynamics, we believe that we need to use internal FRAP to assess the physical state of a P-body, yet our system is not capable of performing a bleaching event in such a small region. We currently have a grant application in review, where we propose such a study and indicate a need to acquire a FRAP module in order to carry out the experiments.

We added the following text (page #9):

" To further investigate the differences in their dynamics, we employed fluorescence recovery after photobleaching (FRAP). We analyzed the colocalization of Me31B-GFP condensates with ERES in small areas of stage 7 egg chambers. After bleaching the area, we allowed for fluorescence recovery, reimaged the region, and recalculated the colocalization (Fig. EV2A). Interestingly, we observed that after photobleaching, there were 57% more cytoplasmic P-bodies in the field of view (Fig. EV2B). This suggests that cytoplasmic P-bodies migrated into the bleached area while ERES-associated P-bodies remained relatively stationary."

Figure 2

12) In Figure 2A, two panels show 90 seconds.

One panel was removed.

13) Figure 2C appears to be reversed with cytoplasm less linear than ERES associated.

This was a big oversight on our part, and we corrected the labeling.

14) The statement, "...suggesting that P-bodies move along microtubules via the Egalitarian/BicD/Dynein complex (38)." - is this referring to in the nurse cells or the oocyte? It has also been previously shown.

While we know it has been shown that P-bodies in general use this mechanism, we intended to highlight that cytoplasmic P-bodies use it while ERES P-bodies do not. We reworded the statement (page #9):

"To determine if the movement of cytoplasmic P-bodies, and not of ERES-associated P-bodies, relied on their association with microtubules, we knocked down BicD using RNAi and assessed Me31B-GFP dynamics (Fig. 2E)."

15) Could cytoplasmic P-bodies be "building" the ERES P-bodies? Is it possible to show that MTs are pulling out cytoplasmic P-bodies versus MTs bringing in cytoplasmic P-bodies. Could this be discussed? (Live imaging showing budding or fusion would help to clarify this, but is not required.)

Throughout our live imaging experiments, we did not visualize any fusion at ERES, but rather only fission or budding off (Figure 3A), leading us to conclude that they are being removed by microtubules. Indeed, we cannot adequately demonstrate that they are being 'pulled off'. We have reworded the section (page #10) to reflect our observations more accurately. Additionally, we expanded Figure 2 data into an additional figure (Figure 3) enabling us to bring more emphasis on alternate explanations. Additionally, we quantified the track duration of colocalized ERES and P-bodies and found their interaction to be long term (Figure EV2D).

16) Is Figure 2H necessary since Figure 2J effectively shows the same data?

While we agree that it shows similar data, we chose to move Figure 2H to supplemental Figure EV2F.

Figure 3

17) Could blocking with RNAi lines cause proteins to accumulate in the ER and lead to increased P-body size? If this is possible perhaps it could be discussed.

It is possible that knocking down COPII components could lead to proteins accumulating at the ER and that this could be the cause of P-body alterations, however we do not believe that this is the case as knockdown of Sec16 and Sar1, the most important components governing ERES formation, resulted in smaller or no P-bodies. We have added a more thorough discussion to the manuscript.

18) Why was anti-CUP used for image and Cup-YFP used for analysis (Figure 3B compared to 3H and 3I).

(Now Figures 4B and 4I, EV3E)

We wanted to perform all our analysis on single fluorescent protein lines for several reasons. As all these proteins colocalize, we wanted to be assured that we were only detecting signal from our protein of interest, however, more importantly, since fluorescent protein tags can sometimes self-aggregate and create a phenotype, we were concerned with performing quantification analysis on egg chambers expressing three fluorescently labeled proteins, as it could result in unreliable data. Therefore, we chose to present an image acquired with two fluorescently labeled proteins and one detected via antibody such that all three proteins could more precisely be visualized together.

19) In Figure 3B, anti-Cup appears in the nucleus and there is a significant amount of Cup detected without Me31B and Tral. Does this suggest that Cup might not be in these P-bodies?

We believe the nuclear puncta may be artifacts of the antibody as we do not detect these with fluorescently labeled proteins. However, we do believe that there are P-bodies with different compositions which are likely performing different functions.

Our response to a similar point raised by Referee #1:

–Assessing the differential composition of P-bodies within an egg chamber has become a significant focus in the field. However, biochemical approaches have not yet succeeded in elucidating any distinct compositions. To address this gap, we plan to undertake a large-scale multiplex imaging screen using endogenously labeled P-body factors in conjunction with multiple maternal mRNAs, which will provide us with a comprehensive understanding of the P-body population. This is however outside the scope of this paper.

20) Figure 1 and 2 shows nicely that there are two populations of P-bodies. In Figure 3B - 3I, the images and analysis appear to be on all P-bodies in these cells. Could this impact the analysis and conclusions? If possible, please comment on this.

We added the following text to convey our reasoning for performing the analysis on all P-bodies (page#12):

“Interestingly, in these knockdown backgrounds, all P-bodies appeared to be similarly affected, despite ERES-P-bodies constituting only ~20% of the total population. This further bolsters the idea that ERES serve as hubs for P-body formation. Given this observation, we decided to quantify all nurse cell P-bodies to acquire a holistic view of the effects of COPII knockdowns on the global P-body population.”

Figure 4

21) The statement, "As many maternal mRNA's are recruited into P-bodies...", needs a reference.

Reference has been added.

Weil, T.T., Parton, R.M., Herpers, B., Soetaert, J., Veenendaal, T., Xanthakis, D., Dobbie, I.M., Halstead, J.M., Hayashi, R., Rabouille, C., Davis, I., 2012. *Drosophila* patterning is established by differential association of mRNAs with P bodies. *Nat Cell Biol* 14, 1305–1313.

22) In Figure 4E, the egg chambers appear to be different stages based on the nucleus position in the oocyte.

They have been replaced with new panels to better match the developmental stages of the egg chambers imaged.

23) Figure 4F does not seem central to the story and could be moved to EV4. It would also be helpful to show a stressed, mcherry^{RNAi} control (or any non-sar1^{RNAi} control).

We respectfully disagree on the importance of 4F and have left the figure as 5K. We do agree that there should be a stressed non-sar1 control, which we now added (Figure 5K).

24) In Figure EV4E and F, P-bodies (marked with Me31B and Cup) look very different. It would be helpful to comment on this.

Yes, these are different images as the zoom is different in order to show different patterns of osk mRNA particles. We have clarified this in the figure legends (now Figure 6H and EV5D).

25) In Figure EV4E and F, the osk mRNA distribution appears different. (E looks good, F needs to be consistent).

This has been reworded as they are showing different zooms.

Fig EV4E (now 6H) was showing a larger area of nurse cells with many P-bodies while EV4F (now EV5D) is showing a zoomed individual P-body. This has been clarified in the legend.

26) In Figure EV4I, it is surprising that Osk protein forms large and distinct particles in the oocyte and is not translated in the nurse cells. Could this be discussed?

Yes, we agree that this expression discrepancy warrants a more extensive discussion.

We moved the results into a main figure (Fig 6A) permitting further commentary. We added the following text (page #21):

“Interestingly, we only observed premature expression of Oskar protein in the oocyte. We favor a model where translational repression can be sufficiently maintained in the nurse cells within the oskar mRNP, although this is not sufficient for fully maintaining oskar mRNA stability without recruitment into P-bodies. In the oocyte, however, where oskar mRNA is highly enriched, the extra layer of protection in P-bodies from the translational machinery is needed to maintain adequate translational repression. We are aware of the limitations of our imaging systems; therefore, we do not overlook the possibility that there is a threshold of protein expression below which we might not be able to detect. Overall, this data suggests that P-bodies are essential for optimal efficiency but are not indispensable to translational repression and RNA stability, which agrees with prior studies (Blake et al., 2024; Eulalio et al., 2007b).”

Referee #3:

The manuscript by Milano et al describes a role for ERES proteins in the localization, biogenesis and/or maintenance of P-bodies. The work is in general well written, and the experiments are conclusive. I think the topic is timely, given the notion that LLPS regulates the formation of ERES and other organelles of the secretory pathways. Therefore, I am generally in favor of publication of this work, but I have some questions and comments that should be addressed before I can fully recommend this paper for publication.

1- My main concern is with the experiments in Figure 3. Firstly, the authors use Sar1 knockdown to disrupt ERES. This is correct, but a knockdown of Sec16 should also disrupt ERES formation. Sec16 is the main organizer of ERES biogenesis. In Drosophila, ERES form on cup-shaped structures and Sec16 is an organizer of these structures. What is the reason for the difference between Sar1 and Sec16? Is it that Sar1 plays a much more fundamental role in P-body formation than Sec16 and other ERES components? I think the authors should either address this experimentally or offer some balanced discussion on this matter. In any case, they should avoid making this distinction between Sar1 and Sec16. The distinction is correct for the more downstream elements such as Sec23, Sec13 and Sec31. These do not affect the formation of ERES, which explains their milder effects.

An alternative interpretation of these results might be that Sec16 per se (and not the ERES) might play an important role in attaching P-bodies to ERES and that Sar1 plays an even more fundamental role. This more fundamental role of Sar1 might also explain why its knockdown affects even the cytoplasmic P-bodies (those not associated with the ERES). My view might well be wrong, but then the authors should explain why. Without such explanation, I am (as a reader) left with too much uncertainty.

Thank you for pointing out the need to make the distinction between the roles of Sec16, Sar1, and the other ERES components, as we agree that Sec16 is fundamental for ERES formation like Sar1. We chose to pursue our studies predominantly with the Sar1 knockdown fly line because it provided a more robust knockdown than that available for Sec16. To further highlight the difference between Sec16 and the coat proteins, we separated the image panels (new Figure 4I). We further added this text (page #13):

“Sec16 is an important protein for dictating ERES localization and thus COPII vesicle formation initiation (Sprangers et al; 2016). The only Sec16 knockdown line available provides a ~60% knockdown (Fig. EV4D). This allowed us to quantify P-bodies in a background of compromised, but not ablated, COPII vesicle formation (Fig. 4I).”

In our proposed mechanism, formation of P-bodies is initiated at ERES, therefore in both the Sec16 and Sar1 knockdowns we see a global effect on P-bodies. Preventing or disrupting the P-bodies at the ERES affects the cytoplasmic P-body population as well. Concurrently, if ERES-associated P-bodies would not precede the formation of cytoplasmic P-bodies, then we wouldn't observe such an effect when ERES are destabilized. It is also possible that Sec16 plays an important role in connecting P-bodies to ERES, especially considering that Sec16 has been shown to undergo LLPS. Nevertheless, it would be difficult to validate such a possibility since it is impossible to assign such a function to Sec16 without disrupting ERES as a whole.

2- Is there a reason why the authors decided not to use a FRAP assay to assess the liquidity of the P-bodies? Unless I am missing something, I think that a FRAP assay will inform more accurately about the liquid state of the P-body than the roughness of its surface. I also think that this is a simple experiment that could be easily done in the course of a revision.

Thank you for raising this point as has Referee # 1.

To assess liquidity of a condensate, an internal FRAP set up is required (single point bleaching) to distinguish 'liquidity' of one P body. Unfortunately, we do not have access to such a system; yet if we had, we would be able to determine only the recovery of a P-body at the ERES. Cytoplasmic P bodies are too mobile and difficult to assess 'recovery' within the volume of a P-body, as movement in and out of the focal plane/frame, makes it a challenging experiment to carry out and interpret.

However, we did employ a modified version of FRAP to assess overall condensate dynamics and added the following text (page #9):

"To further investigate the differences in their dynamics, we employed fluorescence recovery after photobleaching (FRAP). We analyzed the colocalization of Me31B-GFP condensates with ERES in small areas of stage 7 egg chambers. After bleaching the area, we allowed for fluorescence recovery, reimaged the region, and recalculated the colocalization (Fig. EV2A). Interestingly, we observed that after photobleaching, there were 57% more cytoplasmic P-bodies in the field of view (Fig. EV2B). This suggests that cytoplasmic P-bodies migrated into the bleached area while ERES-associated P-bodies remained relatively stationary."

3- Could the authors elaborate a little bit on how they think that microtubules would "pull" on a P-body. An effect of microtubules on motility of P-bodies might be the consequence of a change in cytoplasmic viscosity. This is not a request for an experiment, but I think it would be nice for the reader to read some discussion on this matter.

We distributed the data from previous Figure 2 into two figures (new Figure 2 and 3) to allow for a more thorough discussion of possible causes for the change in P-body distribution. We also added the following text (page #10):

"Microtubule expression was unaffected in the $bicD^{RNAi}$ egg chambers so alterations in cytoplasmic crowding or viscosity did not affect our results (Fig. EV3B)."

Minor comments:

- It would help the general reader if the authors could point out examples for ER-associated and cytoplasmic P-bodies.

Both arrows and arrowheads were added where applicable (Figure 1B,E and Figure 3B,E) to point to these two distinct populations.

- It is also not very clear how the authors determined whether a given P-body is ER-associated.

We added a more thorough explanation of our colocalization analysis (page #6):

"To differentiate between cytoplasmic P-bodies and ER-associated P-bodies we used Imaris image analysis software to detect P-body and ER surfaces. We binned all detected P-bodies that have shortest distances to the ER $\leq 0 \mu\text{m}$ (touch or overlap) as ER-associated P-bodies and any P-bodies with a shortest distance to the ER $> 0 \mu\text{m}$ as a cytoplasmic P-body."

- It would be interesting to report on the instantaneous speed of P-bodies in addition to the MSD and the linearity of the track. I would imagine that cytoplasmic P-bodies are faster because they are less obstructed? The speed cannot be calculated by dividing the distance covered by time. It should be the instantaneous speed that is measured.

We performed the analysis and included this data in Figure 2D.

Dear Prof. Bratu

Thank you for the submission of your revised manuscript to EMBO reports. It has been evaluated by former Referee #3 who now supports publication.

Browsing through the manuscript myself, I noticed a few editorial things that we need before we can proceed with the official acceptance of your study.

- In the figures, the figure panels should be labeled with the letter only, i.e., A instead of A. or B instead of B.
 - Please provide up to 5 keywords
 - Regarding the Author Contributions, we now use CRediT to specify the contributions of each author in the journal submission system. Therefore, please remove the Author Contributions from the manuscript file and make sure that the author contributions in our online manuscript tracking system are correct and up-to-date. The information you specified in the system will be automatically retrieved and typeset into the article. You can enter additional information in the free text box provided, if you wish.
 - References: et al needs to be used after 10 author names, year should be in brackets.
 - Please provide the EV figures as separate, individual figure files instead of the merged PDF.
 - Please provide a legend for Table EV1 and EV2 within a separate tab in each .xls file.
 - Please add an additional callout to Table EV1 in the methods section, where appropriate, so that the table is not only called out in the Reagents and Tools table. You might have this callout already in the methods section "RNA isolation and RT-qPCR", but this callout is to "EV1 Dataset", which doesn't exist. Please correct this.
 - We need the Reagents and Tools table as separate file only. Please remove the table from the manuscript file. Our typesetters will use the Reagent and Tools table file you provided and insert it during the process of typesetting.
 - Please provide the synopsis image in .JPEG, .TIFF or .PNG format. Also note that the final size is 550 pixels wide (and 200-600 pixels high). This is rather small, please make sure that the text is readable at this size (and 100% zoom on the screen).
 - Please remove the synopsis text from the manuscript file and upload it as separate document.
 - Source data: please sort the source data within a figure folder into subfolders for each panel.
 - Moreover, we need separate .xls files per each figure panel.
 - A spot checked showed that the source data for Figure 1 is mislabeled. 1A should be 1B; 1D should be 1E; 1G should be 1E. Please check.
 - Please enter the following fund also in the online manuscript tracking system: NIH/NIGMS RO1-GM084947.
 - Please provide figure titles for Figures EV 1-5 in the figure legends. Please also add the headers 'Figure legends' and 'Expanded View figure legends'.
 - Our production/data editors have asked you to clarify several points in the figure legends (see below). Please incorporate these changes in the manuscript and return the revised file with tracked changes with your final manuscript submission.
- A) Replicates and error bars:
- Please note that the measure of center for the error bars needs to be defined in the legends of figures EV 4a; EV 5b-c.
- B) Data presentation:
- Please note that the scale bar needs to be defined for figures 2a-b, e.
 - Please note that the white arrowheads/arrows are not defined in the legends of figures 2a-b, e; EV 2a. This needs to be rectified.
 - Author Checklist "Design - Laboratory protocol" (row 77): you indicate that this information is included in the manuscript. Please note that this point would only refer to the use or reference of external step-by-step protocols that were e.g. published on protocols.io or in a methods journal. Please check this statement again.
 - Author Checklist "Experimental study design and statistics - Were any steps taken to minimize the effects of subjective bias ..."

(row 81). You indicate that you included such information in the methods section, but I could not locate it. Please either include a statement on randomization in the methods or change you answer to this point in the checklist.

- As a standard procedure, we edit the title and abstract of manuscripts to make them more accessible to a general readership. Please find my proposal below my signature.

Kind regards,

Martina

=====

ER exit sites contribute to P-body organization and integrity during *Drosophila melanogaster* oogenesis

Processing bodies (P-bodies) are cytoplasmic membrane-less organelles which host multiple mRNA processing events. While the fundamental principles of P-body organization are beginning to be elucidated in vitro, a nuanced understanding of how their assembly is regulated in vivo remains elusive. Here, we investigate the potential link between ER exit sites and P-bodies in *Drosophila melanogaster* egg chambers. Employing a combination of live and super-resolution imaging, we find that P-bodies associated with ER exit sites are larger and less mobile than cytoplasmic P-bodies, indicating that they constitute a distinct class of P-bodies. Moreover, we demonstrate that altering the composition of ER exit sites has differential effects on core P-body proteins (Me31B, Cup, and Trailer Hitch), suggesting a potential role for ER exit sites in P-body organization. Furthermore, we show that in the absence of ER exit sites, P-body integrity is compromised and the stability and translational repression efficiency of the maternal mRNA, oskar, are reduced. Together, our data highlights the crucial role of ER exit sites in governing P-body organization.

=====

Referee #3:

The authors have responded to all my initial comments and concerns. I have no further points to raise and think that the manuscript is now ready for publication.

All editorial and formatting issues were resolved by the authors.

Prof. Diana Bratu
Hunter College
695 Park Ave.
HN914
New York, NY 10065-5024
United States

Dear Prof. Bratu,

I am very pleased to accept your manuscript for publication in the next available issue of EMBO reports. Thank you for your contribution to our journal.

Kind regards,
